# Entropic Neural Optimal Transport via Diffusion Processes

**Nikita Gushchin**
Skoltech*
*Moscow, Russia*
`n.gushchin@skoltech.ru`

**Alexander Kolesov**
Skoltech*
*Moscow, Russia*
`a.kolesov@skoltech.ru`

**Alexander Korotin**
Skoltech*, AIRI[†]
*Moscow, Russia*
`a.korotin@skoltech.ru`

**Dmitry Vetrov**
HSE University[‡], AIRI[†]
*Moscow, Russia*
`vetrovd@yandex.ru`

**Evgeny Burnaev**
Skoltech*, AIRI[†]
*Moscow, Russia*
`e.burnaev@skoltech.ru`

## Abstract

We propose a novel neural algorithm for the fundamental problem of computing the entropic optimal transport (EOT) plan between continuous probability distributions which are accessible by samples. Our algorithm is based on the saddle point reformulation of the dynamic version of EOT which is known as the Schrödinger Bridge problem. In contrast to the prior methods for large-scale EOT, our algorithm is end-to-end and consists of a single learning step, has fast inference procedure, and allows handling small values of the entropy regularization coefficient which is of particular importance in some applied problems. Empirically, we show the performance of the method on several large-scale EOT tasks. The code for the ENOT solver can be found at `https://github.com/ngushchin/EntropicNeuralOptimalTransport`.

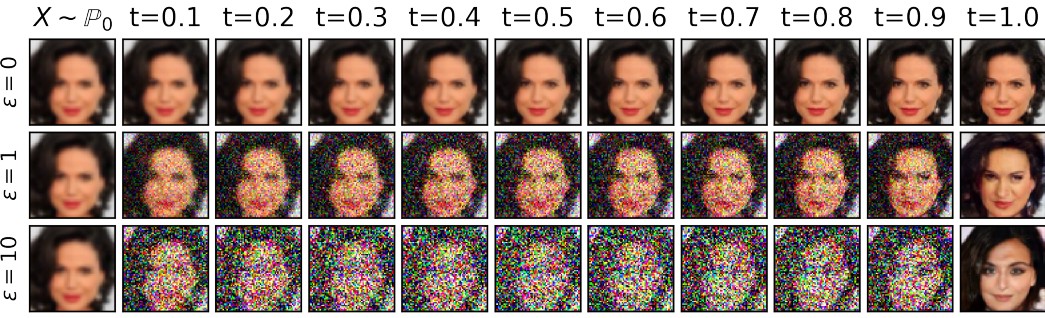

Figure 1: Trajectories of samples learned by our Algorithm 1 for Celeba deblurring with $\epsilon = 0, 1, 10$.

## 1 Introduction

Optimal transport (OT) plans are a fundamental family of alignments between probability distributions. The majority of scalable neural algorithms to compute OT are based on the dual formulations of OT, see [28] for a survey. Despite the success of such formulations in generative modeling [42, 31], these dual form approaches can hardly be generalized to the popular **entropic OT** [12]. This is due to the numerical instability of the dual entropic OT problem [14] which appears for small entropy

---

*Skolkovo Institute of Science and Technology

[†]Artificial Intelligence Research Institute

[‡]HSE University

37th Conference on Neural Information Processing Systems (NeurIPS 2023).

regularization values which are suitable for downstream generative modeling tasks. At the same time, entropic OT is useful as it allows to learn one-to-many stochastic mappings with tunable level of sample diversity. This is particularly important for ill-posed problems such as super-resolution [37].

**Contributions**. We propose a saddle-point reformulation of the Entropic OT problem via using its dynamic counterpart known as the Schrödinger Bridge problem (§4.1). Based on our new reformulation, we propose a novel end-to-end neural algorithm to solve the related entropic OT problem for a pair of continuous distributions accessible by samples (§4.2). Unlike many predecessors, our method allows handling small entropy coefficients. This enables practical applications to *data → data* mapping tasks that require slight variability in the learned maps. Furthermore, we provide an error analysis for solving the suggested saddle point optimization problem through duality gaps which are the errors of solving the inner and outer optimization problems (§4.3). Empirically, we illustrate the performance of the method on several toy and large-scale EOT tasks (§5).

## 2 Background

Optimal Transport (OT) and Schrödinger Bridge (SB) problems imply finding an efficient way to transform some initial distribution $\mathbb{P}_0$ to target distribution $\mathbb{P}_1$. While the solution of OT only gives the information about which part of $\mathbb{P}_0$ is transformed to which part of $\mathbb{P}_1$, SB implies finding a stochastic process that describes the entire evolution from $\mathbb{P}_0$ to $\mathbb{P}_1$. Below we give an introduction to OT and SB problems and show how they are related. For a detailed overview of OT, we refer to [49, 41] and of SB – to [32, 11].

### 2.1 Optimal Transport (OT)

**Kantorovich's OT formulation** (with the quadratic cost). We consider $D$-dimensional Euclidean spaces $\mathcal{X}$, $\mathcal{Y}$ and use $\mathcal{P}_2(\mathcal{X}) = \mathcal{P}_2(\mathcal{Y})$ to denote the respective sets of Borel probability distributions on them which have finite second moment. For two distributions $\mathbb{P}_0 \in \mathcal{P}_2(\mathcal{X})$, $\mathbb{P}_1 \in \mathcal{P}_2(\mathcal{Y})$, consider the following minimization problem:

$$\inf_{\pi \in \Pi(\mathbb{P}_0, \mathbb{P}_1)} \int_{\mathcal{X} \times \mathcal{Y}} \frac{||x - y||^2}{2} d\pi(x, y), \tag{1}$$

where $\Pi(\mathbb{P}_0, \mathbb{P}_1) \subset \mathcal{P}_2(\mathcal{X} \times \mathcal{Y})$ is the set of probability distributions on $\mathcal{X} \times \mathcal{Y}$ with marginals $\mathbb{P}_0$ and $\mathbb{P}_1$. Such distributions $\pi \in \Pi(\mathbb{P}_0, \mathbb{P}_1)$ are called the transport plans between $\mathbb{P}_0$ and $\mathbb{P}_1$. The set $\Pi(\mathbb{P}_0, \mathbb{P}_1)$ is non-empty as it always contains the trivial plan $\mathbb{P}_0 \times \mathbb{P}_1$. A minimizer $\pi^*$ of (1) always exists and is called an OT plan. If $\mathbb{P}_0$ is absolutely continuous, then $\pi^*$ is deterministic: its conditional distributions are degenerate, i.e., $\pi^*(\cdot|x) = \delta_{T^*(x)}$ for some $T^* : \mathcal{X} \to \mathcal{Y}$ (OT map).

**Entropic OT formulation.** We use $H(\pi)$ to denote the differential entropy of distribution $\pi$ and $\text{KL}(\pi||\pi')$ to denote the Kullback-Leibler divergence between distributions $\pi$ and $\pi'$. Two most popular entropic OT formulations regularize (1) with the entropy $H(\pi)$ or KL-divergence between plan $\pi$ and the trivial plan $\mathbb{P}_0 \times \mathbb{P}_1$, respectively ($\epsilon > 0$):

$$\inf_{\pi \in \Pi(\mathbb{P}_0, \mathbb{P}_1)} \int_{\mathcal{X} \times \mathcal{Y}} \frac{||x - y||^2}{2} d\pi(x, y) - \epsilon H(\pi), \tag{2}$$

$$\inf_{\pi \in \Pi(\mathbb{P}_0, \mathbb{P}_1)} \int_{\mathcal{X} \times \mathcal{Y}} \frac{||x - y||^2}{2} d\pi(x, y) + \epsilon \text{KL}(\pi || \mathbb{P}_0 \times \mathbb{P}_1). \tag{3}$$

Since $\pi \in \Pi(\mathbb{P}_0, \mathbb{P}_1)$, it holds that $\text{KL}(\pi||\mathbb{P}_0 \times \mathbb{P}_1) = -H(\pi) + H(\mathbb{P}_0) + H(\mathbb{P}_1)$, i.e., both formulations are equal up to an additive constant when $\mathbb{P}_0$ and $\mathbb{P}_1$ are absolutely continuous. The minimizer of (2) is **unique** since the functional is strictly convex in $\pi$ thanks to the strict convexity of $H(\pi)$. This unique minimizer $\pi^*$ is called the entropic OT plan.

### 2.2 Schrödinger Bridge (SB)

**SB with the Wiener prior.** Let $\Omega$ be the space of $\mathbb{R}^D$ valued functions of time $t \in [0, 1]$ describing some trajectories in $\mathbb{R}^D$, which start at time $t = 0$ and end at time $t = 1$. We use $\mathcal{P}(\Omega)$ to denote the set of probability distributions on $\Omega$. We use $dW_t$ to denote the differential of the standard Wiener process. Let $W^\epsilon \in \mathcal{P}(\Omega)$ be the Wiener process with the variance $\epsilon$ which starts at $\mathbb{P}_0$. This diffusion process can be represented via the following stochastic differential equation (SDE):

$$W^\epsilon : dX_t = \sqrt{\epsilon} dW_t, \quad X_0 \sim \mathbb{P}_0. \tag{4}$$

We use $\mathrm{KL}(T||Q)$ to denote the Kullback-Leibler divergence between stochastic processes $T$ and $Q$. The Schrödinger Bridge problem was initially proposed in 1931/1932 by Erwin Schrödinger [44]. It can be formulated as follows [11, Problem 4.1]:

$$\inf_{T \in \mathcal{F}(\mathbb{P}_0, \mathbb{P}_1)} \mathrm{KL}(T||W^\epsilon), \tag{5}$$

where $\mathcal{F}(\mathbb{P}_0, \mathbb{P}_1) \subset \mathcal{P}(\Omega)$ is the set of probability distributions on $\Omega$ having marginal distributions $\mathbb{P}_0$ and $\mathbb{P}_1$ at $t = 0$ and $t = 1$, respectively. Thus, the Schrödinger Bridge problem implies finding a stochastic process $T$ with marginal distributions $\mathbb{P}_0$ and $\mathbb{P}_1$ at times $t = 0$ and $t = 1$, respectively, which has minimal KL divergence with the prior process $W^\epsilon$.

**Link to OT problem.** Here we recall how Scrödinger Bridge problem (5) relates to entropic OT problem (2). *This relation is well known (see, e.g., [32] or [11, Problem 4.2]), and we discuss it in detail because our proposed approach (§4) is based on it.* Let $\pi^T$ denote the joint distribution of a stochastic process $T$ at time moments $t = 0, 1$ and $\pi_0^T, \pi_1^T$ denote its marginal distributions at time moments $t = 0, 1$, respectively. Let $T_{|x,y}$ denote the stochastic processes $T$ conditioned on values $x, y$ at times $t = 0, 1$, respectively. One may decompose $\mathrm{KL}(T||W^\epsilon)$ as [48, Appendix C]:

$$\mathrm{KL}(T||W^\epsilon) = \mathrm{KL}(\pi^T||\pi^{W^\epsilon}) + \int_{\mathcal{X} \times \mathcal{Y}} \mathrm{KL}(T_{|x,y}||W_{|x,y}^\epsilon) d\pi^T(x,y), \tag{6}$$

i.e., KL divergence between $T$ and $W^\epsilon$ is a sum of two terms: the first represents the similarity of the processes at start and finish times $t = 0$ and $t = 1$, while the second term represents the similarity of the processes for intermediate times $t \in (0, 1)$ conditioned on the values at $t = 0, 1$. For the first term, it holds (see Appendix A or [11, Eqs 4.7-4.9]):

$$\mathrm{KL}(\pi^T||\pi^{W^\epsilon}) = \int_{\mathcal{X} \times \mathcal{Y}} \frac{||x - y||^2}{2\epsilon} d\pi^T(x,y) - H(\pi^T) + C, \tag{7}$$

where $C$ is a constant which depends only on $\mathbb{P}_0$ and $\epsilon$. In [32, Proposition 2.3], the authors show that if $T^*$ is the solution to (5), then $T_{|x,y}^* = W_{|x,y}^*$. Hence, one may optimize (5) over processes $T$ for which $T_{|x,y} = W_{|x,y}^\epsilon$ for every $x, y$ and set the last term in (6) to zero. In this case:

$$\inf_{T \in \mathcal{F}(\mathbb{P}_0, \mathbb{P}_1)} \mathrm{KL}(T||W^\epsilon) = \inf_{T \in \mathcal{F}(\mathbb{P}_0, \mathbb{P}_1)} \mathrm{KL}(\pi^T||\pi^{W^\epsilon}) = \inf_{\pi^T \in \Pi(\mathbb{P}_0, \mathbb{P}_1)} \mathrm{KL}(\pi^T||\pi^{W^\epsilon}). \tag{8}$$

i.e., it suffices to optimize only over joint distributions $\pi^T$ at time moments $t = 0, 1$. *Hence, minimizing (8) is equivalent (up to an additive constant $C$) to solving EOT (2) divided by $\epsilon$, and their respective solutions $\pi^{T^*}$ and $\pi^*$ coincide.* Thus, SB problem (5) can be simplified to the entropic OT problem (2) with entropy coefficient $\epsilon$.

**Dual form of EOT**. Entropic OT problem (8) has several dual formulations. Here we recall the one which is particularly useful to derive our algorithm. The dual formulation follows from the weak OT theory [7, Theorem 1.3] and we explain it in detail in the proof of Lemma B.3:

$$\inf_{\pi \in \Pi(\mathbb{P}_0, \mathbb{P}_1)} \mathrm{KL}(\pi||\pi^{W^\epsilon}) = \sup_\beta \Big\{ \int_{\mathcal{X}} \beta^C(x) d\mathbb{P}_0(x) + \int_{\mathcal{Y}} \beta(y) d\mathbb{P}_1(y) \Big\},$$

where $\beta^C(x) \stackrel{\text{def}}{=} \inf_{\nu \in \mathcal{P}_2(\mathcal{Y})} \big\{ \mathrm{KL}\big(\nu||\pi^{W^\epsilon}(\cdot|x)\big) - \int_{\mathcal{Y}} \beta(y) d\nu(y) \big\}$. The sup here is taken over $\beta$ belonging to the set of functions

$$\mathcal{C}_{b,2}(\mathcal{Y}) \stackrel{\text{def}}{=} \{\beta : \mathcal{Y} \to \mathbb{R} \text{ continuous s.t. } \exists u, v, w \in \mathbb{R} : u||\cdot||^2 + v \leq \beta(\cdot) \leq w\},$$

i.e., $\beta$ should be continuous with mild boundness assumptions.

**Dynamic SB problem (DSB)**. It is known that the solution to the SB problem (5) belongs to the class $\mathcal{D}(\mathbb{P}_0)$ of finite-energy diffusions $T_f$ [32, Proposition 4.1] which are given by:

$$T_f : dX_t = f(X_t, t)dt + \sqrt{\epsilon}dW_t, \quad X_0 \sim \mathbb{P}_0, \quad \mathbb{E}_{T_f}\Big[\int_0^1 ||f(X_t, t)||^2 dt\Big] < \infty, \tag{9}$$

where $f : \mathbb{R}^D \times [0, 1] \to \mathbb{R}^D$ is the drift function. The last inequality in (9) means that $T_f$ is a finite-energy diffusion. Hence, optimizing only over finite-energy diffusions rather than all possible

processes is enough to solve SB problem (5). For finite-energy diffusions, it is possible to rewrite the optimization objective. One can show that $\mathrm{KL}(T_f||W^\epsilon)$ between processes $T_f$ and $W^\epsilon$ is [40]:

$$\mathrm{KL}(T_f||W^\epsilon) = \frac{1}{2\epsilon}\mathbb{E}_{T_f}[\int_0^1 ||f(X_t, t)||^2 dt]. \tag{10}$$

By substituting (10) in (5), SB problem reduces to the following problem [11, Problem 4.3]:

$$\inf_{T_f \in \mathcal{D}(\mathbb{P}_0, \mathbb{P}_1)} \mathrm{KL}(T_f||W^\epsilon) = \inf_{T_f \in \mathcal{D}(\mathbb{P}_0, \mathbb{P}_1)} \frac{1}{2\epsilon}\mathbb{E}_{T_f}[\int_0^1 ||f(X_t, t)||^2 dt], \tag{11}$$

where $\mathcal{D}(\mathbb{P}_0, \mathbb{P}_1) \subset \mathcal{P}(\Omega)$ is the set of finite-energy diffusion on $\Omega$ having marginal distributions $\mathbb{P}_0$ and $\mathbb{P}_1$ at $t = 0$ and $t = 1$, respectively. Note that $\mathcal{D}(\mathbb{P}_0, \mathbb{P}_1) \subset \mathcal{F}(\mathbb{P}_0, \mathbb{P}_1)$. Since the problem (11) is the equivalent reformulation of (5), its optimal value also equals (8). However, solving this problem, as well as (8) is challenging as it is still hard to satisfy the boundary constraints.

## 3 Related Work

In this section, we overview the existing methods to compute the OT plan or map. To avoid any confusion, we emphasize that popular Wasserstein GANs [5] compute only the OT cost but not the OT plan and, consequently, are out of scope of the discussion.

**Discrete OT.** The majority of algorithms in computational OT are designed for the discrete setting where the inputs $\mathbb{P}_0, \mathbb{P}_1$ have finite supports. In particular, the usage of entropic regularization (2) allows to establish efficient methods [13] to compute the entropic OT plan between discrete distributions with the support size up to $10^5$-$10^6$ points, see [41] for a survey. For larger support sizes, such methods are typically computationally intractable.

**Continuous OT.** Continuous methods imply computing the OT plan between distributions $\mathbb{P}_0$, $\mathbb{P}_1$ which are accessible by empirical samples. Discrete methods "as-is" are not applicable to the continuous setup in high dimensions because they only do a stochastic *matching* between the train samples and do not provide out-of-sample estimation. In contrast, continuous methods employ neural networks to explicitly or implicitly *learn* the OT plan or map. As a result, these methods can map unseen samples from $\mathbb{P}_0$ to $\mathbb{P}_1$ according to the learned OT plan or map.

There exists many methods to compute OT plans [51, 36, 38, 26, 28, 27, 42, 16, 19, 18] but they consider only unregularized OT (1) rather than the entropic one (2). In particular, they mostly focus on computing the deterministic OT plan (map) which may not exist. Recent works [31, 30] design algorithms to compute OT plans for weak OT [20, 7]. Although weak OT technically subsumes entropic OT , these works do not cover the entropic OT (2) because there is no simple way to estimate the entropy from samples. Below we discuss methods specifically for **EOT** (2) and **DSB** (11).

### 3.1 Continuous Entropic OT

In LSOT [45], the authors solve the dual problem to entropic OT (2). The dual potentials are then used to compute the *barycentric projection* $x \mapsto \int_{\mathcal{Y}} y \, d\pi^*(y|x)$, i.e., the first conditional moment of the entropic OT plan. This strategy may yield a deterministic approximation of $\pi^*$ for small $\epsilon$ but does not recover the entire plan itself.

In [14, Figure 3], the authors show that the barycentric projection leads to the averaging artifacts which make it impractical in downstream tasks such as the unpaired image super-resolution. To solve these issues, the authors propose a method called SCONES. It recovers the entire conditional distribution $\pi^*(y|x)$ of the OT plan $\pi^*$ from the dual potentials. Unfortunately, this is costly. During the training phase, the method requires learning a score-based model for the distribution $\mathbb{P}_1$. More importantly, during the inference phase, one has to run the Langevin dynamic to sample from $\pi^*(y|x)$.

The optimization of the above-mentioned entropic approaches requires evaluating the exponent of large values which are proportional to $\epsilon^{-1}$ [45, Eq. 7]. Due to this fact, those methods are **unstable** for small values $\epsilon$ in (2). In contrast, our proposed method (§4) resolves this issue: technically, it works even for $\epsilon = 0$.

### 3.2 Approaches to Compute Schrödinger Bridges

Existing approaches to solve DSB mainly focus on generative modeling applications (*noise → data*). For example, FB-SDE [10] utilizes data likelihood maximization to optimize the parameters of

forward and backward SDEs for learning the bridge. On the other hand, MLE-SB [48] and DiffSB [15] employ the iterative proportional fitting technique to learn the DSB. Another method proposed by [50] involves solving the Schrodinger Bridge only between the Dirac delta distribution and real data to solve the data generation problem.

Recent studies [34, 33] have indicated that the Schrödinger Bridge can be considered as a specific instance of a more general problem known as the Mean-Field Game [1]. These studies have also suggested novel algorithms for resolving the Mean-Field Game problem. However, the approach presented by [33] cannot be directly applied to address the hard distribution constraints imposed on the start and final probability distribution as in the Schrödinger Bridge problem (see Appendix H). The approach suggested by [34] coincides with that proposed in FB-SDE [10] for the SB problem.

## 4 The Algorithm

This section presents our novel neural network-based algorithm to recover the solution $T_{f*}$ of the DSB problem (11) and the solution $\pi^*$ of the related EOT problem (8). In §4.1, we theoretically derive the proposed saddle point optimization objective. In §4.2 we provide and describe the practical learning procedure to optimize it. In §4.3, we perform the error analysis via duality gaps. In Appendix B, we provide proofs of all theorems and lemmas.

### 4.1 Saddle Point Reformulation of EOT via DSB

For two distributions $\mathbb{P}_0$ and $\mathbb{P}_1$ accessible by finite empirical samples, solving entropic OT (2) "as-is" is non-trivial. Indeed, one has to **(a)** enforce the marginal constraints $\pi \in \Pi(\mathbb{P}_0, \mathbb{P}_1)$ and **(b)** estimate the entropy $H(\pi)$ from empirical samples which is challenging. Our idea below is to employ the relation of EOT with DSB to derive an optimization objective which in practice can recover the entropic plan avoiding the above-mentioned issues. First, we introduce the functional

$$\mathcal{L}(\beta, T_f) \stackrel{\text{def}}{=} \left\{ \overbrace{\mathbb{E}_{T_f}\left[\frac{1}{2\epsilon}\int_0^1 ||f(X_t, t)||^2 dt\right]}^{=\text{KL}(T_f||W^\epsilon)} - \int_{\mathcal{Y}} \beta(y) d\pi_1^{T_f}(y) + \int_{\mathcal{Y}} \beta(y) d\mathbb{P}_1(y) \right\}. \quad (12)$$

This functional can be viewed as the Lagrangian for DSB (11) with the relaxed constraint $d\pi_1^{T_f}(y) = d\mathbb{P}_1(y)$, and function $\beta : \mathcal{Y} \to \mathbb{R}$ (*potential*) playing the role of the Langrange multiplier.

**Theorem 4.1** (Relaxed DSB formulation). *Consider the following saddle point optimization problem:*

$$\sup_\beta \inf_{T_f} \mathcal{L}(\beta, T_f) \quad (13)$$

*where* sup *is taken over potentials* $\beta \in \mathcal{C}_{b,2}(\mathcal{Y})$ *and* inf *is taken over diffusion processes* $T_f \in \mathcal{D}(\mathbb{P}_0)$. *Then for every optimal pair* $(\beta^*, T_{f*})$ *for (13), i.e.,*

$$\beta^* \in \operatorname*{argsup}_\beta \inf_{T_f} \mathcal{L}(\beta, T_f) \quad \text{and} \quad T_{f*} \in \operatorname*{arginf}_{T_f} \mathcal{L}(\beta^*, T_f),$$

*it holds that the process* $T_f^*$ *is the solution to SB (11).*

**Corollary 4.2** (Entropic OT as relaxed DSB). *If* $(\beta^*, T_{f*})$ *solves* (13), *then* $\pi^{T_{f*}}$ *is the EOT plan (2).*

Our results above show that by solving (13), one immediately recovers the optimal process $T_{f*}$ in DSB (11) and the optimal EOT plan $\pi^* = \pi^{T_f^*}$. The notable benefits of considering (13) instead of (2), (3) and (11) is that **(a)** it is as an optimization problem over $(\beta, T_f)$ without the constraint $d\pi_1^{T_f}(y) = d\mathbb{P}_1(y)$, and **(b)** objective (12) admits Monte-Carlo estimates by using random samples from $\mathbb{P}_0, T_f, \mathbb{P}_1$. In §4.2 below, we describe the straightforward practical procedure to optimize (13) with stochastic gradient methods and neural nets.

**Relation to prior works.** In the field of neural OT, there exist so many maximin reformulations of OT (classic [28, 42, 16, 19, 22], weak [31, 30], general [6]) resembling our (13) that a reader may naturally wonder **(1)** why not to apply them to solve entropic OT? **(2)** how does our reformulation differ from all of them? It is indeed true that, e.g., algorithm from [31, 6] mathematically covers the entropic case (2). Yet in practice it requires estimation of entropy from samples for which there is no easy way and the authors do not consider this case. These methods can be hardly applied to EOT.

In contrast to the prior works, we deal with entropic OT through its connection with the Schrödinger Bridge. Our max-min reformulation is for DSB and it avoids computing the entropy term $H(\pi)$

in EOT. This term is replaced by the energy of the process $\mathbb{E}_{T_f}\big[\int_0^1 ||f(X_t,t)||^2 dt\big]$ which can be straightforwardly estimated from the samples of $T_f$, allowing to establish a computational algorithm.

## 4.2 Practical Optimization Procedure

To solve (12), we parametrize drift function $f(x,t)$ of the process $T_f$ and potential $\beta(y)$[4] by neural nets $f_\theta : \mathbb{R}^D \times [0,1] \to \mathbb{R}^D$ and $\beta_\phi : \mathbb{R}^D \to \mathbb{R}$. We consider the following maximin problem:

$$\sup_\beta \inf_{T_{f_\theta}} \Big\{ \frac{1}{2\epsilon}\mathbb{E}_{T_{f_\theta}}\big[\int_0^1 ||f_\theta(X_t,t)||^2 dt\big] + \int_{\mathcal{Y}} \beta_\phi(y)d\mathbb{P}_1(y) - \int_{\mathcal{Y}} \beta_\phi(y)d\pi_1^{T_{f_\theta}}(y)\Big\}. \quad (14)$$

We use standard Euler-Maruyama (Eul-Mar) simulation (Algorithm 2 in Appendix C) for sampling from the stochastic process $T_f$ by solving its SDE (9). To estimate the value of $\mathbb{E}_{T_f}[\int_0^1 ||f(X_t,t)||^2 dt]$ in (14), we utilize the mean value of $||f(x,t)||^2$ over time $t$ of trajectory $X_t$ that is obtained during the simulation by Euler-Maruyama algorithm (Appendix C). We train $f_\theta$ and $\beta_\phi$ by optimizing (12) with the stochastic gradient ascent-descent by sampling random batches from $\mathbb{P}_0$, $\mathbb{P}_1$. The optimization procedure is detailed in Algorithm 1. We use $f_{n,m}$ to denote the drift at time step $n$ for the $m$-th object of the input sample batch. We use the averaged of the drifts as an estimate of $\int_0^1 ||f(X_t,t)||^2 dt$ in the training objective.

**Remark.** For the image tasks (§5.3, §5.4), we find out that using a slightly different parametrization of $T_f$ considerably improves the quality of our Algorithm, see Appendix F.

It should be noted that the term $\mathbb{E}_{T_f}[\int_0^1 ||f(X_t,t)||^2 dt]$ is not multiplied by $\frac{1}{2\epsilon}$ in the algorithm since this does not affect the optimal $T_{f^*}$ solving the inner optimization problem, see Appendix D.

---

**Algorithm 1:** Entropic Neural OT (ENOT)

**Input:**
  samples from distributions $\mathbb{P}_0$, $\mathbb{P}_1$;
  Wiener prior noise variance $\epsilon \geq 0$;
  drift network $f_\theta : \mathbb{R}^D \times [0,1] \to \mathbb{R}^D$;
  beta network $\beta_\phi : \mathbb{R}^D \to \mathbb{R}$;
  number of steps $N$ for Eul-Mar (App C);
  number of inner iterations $K_f$.

**Output:** drift $f_\theta^*$ of $T_{f_\theta^*}$ solving DSB (11).

**repeat**
  Sample batches $X_0 \sim \mathbb{P}_0$, $Y \sim \mathbb{P}_1$;
  $\{X_n, f_n\}_{n=0}^N \leftarrow$ Eul-Mar$(X_0, T_{f_\theta})$;
  $\mathcal{L}_\beta \leftarrow \frac{1}{|X_N|}\sum\limits_{x \in X_N} \beta_\phi(x) - \frac{1}{|Y|}\sum\limits_{y \in Y} \beta_\phi(y)$;
  Update $\phi$ by using $\frac{\partial L_\beta}{\partial \phi}$ ;
  **for** $k=1$ **to** $K_f$ **do**
    Sample batches $X_0 \sim \mathbb{P}_0$, $Y \sim \mathbb{P}_1$;
    $\{X_n, f_n\}_{n=0}^N \leftarrow$ Eul-Mar$(X_0, T_{f_\theta})$;
    $\widehat{\mathrm{KL}} \leftarrow \frac{1}{N}\sum\limits_{n=0}^{N-1} \frac{1}{|f_n|}\sum\limits_{m=1}^{|f_n|} ||f_{n,m}||^2$ ;
    $\mathcal{L}_f \leftarrow \widehat{\mathrm{KL}} - \frac{1}{|X_N|}\sum\limits_{x \in X_N} \beta_\phi(x)$;
    Update $\theta$ by using $\frac{\partial L_\theta}{\partial \theta}$;

**until** *converged*;

---

**Relation to GANs.** At the first glance, our method might look like a typical GAN as it solves a maximin problem with the "discriminator" $\beta_\phi$ and SDE "generator" with the drift $f_\theta$. Unlike GANs, in our saddle point objective (12), optimization of "generator" $T_f$ and "discriminator" $\beta$ are swapped, i.e., "generator" is adversarial to "discriminator", not vise versa, as in GANs. For further discussion of differences between saddle point objectives of neural OT/GANs, see [31, §4.3], [42, §4.3], [16].

## 4.3 Error Bounds via Duality Gaps

Our algorithm solves a maximin optimization problem and recovers some *approximate* solution $(\hat{\beta}, T_{\hat{f}})$. Given such a pair, it is natural to wonder how close is the recovered $T_{\hat{f}}$ to the optimal $T_{f^*}$. Our next result sheds light on this question via bounding the error with the *duality gaps*.

**Theorem 4.3** (Error analysis via duality gaps). *Consider a pair $(\hat{\beta}, T_{\hat{f}})$. Define the duality gaps, i.e., errors of solving inner and outer optimization problems by:*

$$\epsilon_1 \overset{def}{=} \mathcal{L}(\hat{\beta}, T_{\hat{f}}) - \inf_{T_f} \mathcal{L}(\hat{\beta}, T_f), \qquad and \qquad \epsilon_2 \overset{def}{=} \sup_\beta \inf_{T_f \in \mathcal{D}(\mathbb{P}_0)} \mathcal{L}(\beta, T_f) - \inf_{T_f} \mathcal{L}(\hat{\beta}, T_f). \quad (15)$$

*Then it holds that*

$$\rho_{TV}(T_{\hat{f}}, T_{f^*}) \leq \sqrt{\epsilon_1 + \epsilon_2}, \qquad and \qquad \rho_{TV}(\pi^{T_{\hat{f}}}, \pi^{T_{f^*}}) \leq \sqrt{\epsilon_1 + \epsilon_2}, \quad (16)$$

*where we use $\rho_{TV}(\cdot, \cdot)$ to denote the total variation norm (between the processes or plans).*

---

[4]In practice, $\beta_\phi \in \mathcal{C}_{b,2}(\mathcal{Y})$ since we can choose $u=0$, $v = \min(\texttt{float32})$, $w = \max(\texttt{float32})$.

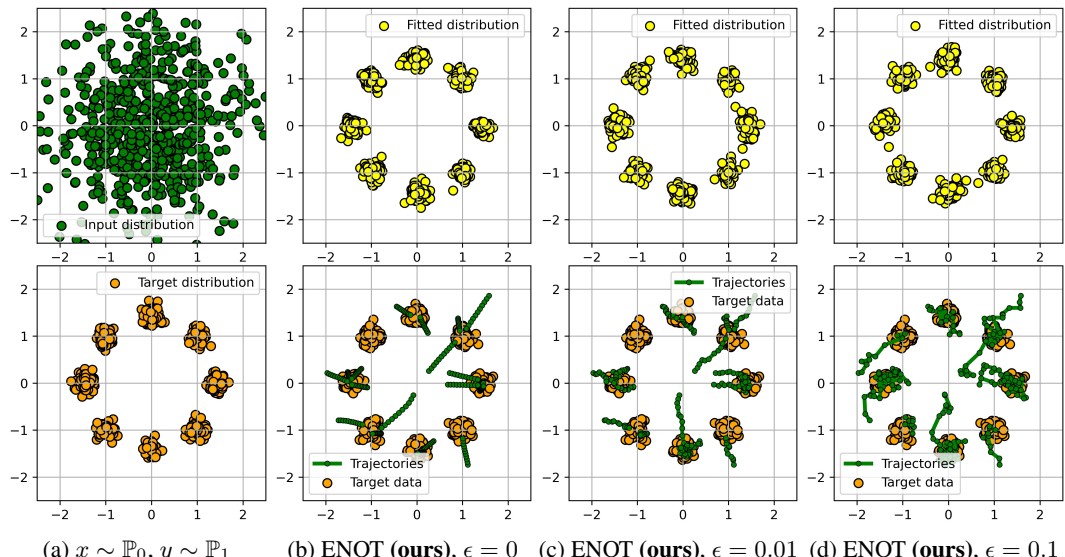

(a) $x \sim \mathbb{P}_0, y \sim \mathbb{P}_1$    (b) ENOT (**ours**), $\epsilon = 0$    (c) ENOT (**ours**), $\epsilon = 0.01$    (d) ENOT (**ours**), $\epsilon = 0.1$

Figure 2: *Gaussian→Mix of 8 Gaussians.* The process learned with ENOT (**ours**) for $\epsilon = 0, 0.01, 0.1$.

**Relation to prior works**. There exist deceptively similar results, see [38, Theorem 3.6], [42, Theorem 4.3], [16, Theorem 4], [6, Theorem 3]. *None of them are relevant to our EOT/DSB case.*

In [38], [42], [16], the authors consider maximin reformulations of unregularized OT (1), i.e., non-entropic. Their result requires the potential $\beta$ ($f, \psi$ in their notation) to be a convex function which in practice means that one has to employ ICNNs [4] which have poor expressiveness [29, 17, 26]. Our result is free from such assumptions on $\beta$. In [6], the authors consider general OT problem [39] and require the general cost functional to be strongly convex (in some norm). Their results also do not apply to our case as the (negative) entropy which we consider is not strongly convex.

## 5 Experimental Illustrations

In this section, we qualitatively and quantitatively illustrate the performance of our algorithm in several entropic OT tasks. Our proofs apply only to EOT ($\epsilon > 0$), but for completeness, we also present results $\epsilon = 0$, i.e., unregularized case (1). Furthermore, we test our algorithm with $\epsilon = 0$ on the Wasserstein-2 Benchmark [28], see Appendix J. We also demonstrate the extension of our algorithm to costs other than the squared Euclidean distance in Appendix I. The implementation details are given in Appendices E, F and G. The code is written in PyTorch and is publicly available at

https://github.com/ngushchin/EntropicNeuralOptimalTransport

### 5.1 Toy 2D experiments

Here we give qualitative examples of our algorithm's performance on toy 2D pairs of distributions. We consider two pairs $\mathbb{P}_0, \mathbb{P}_1$: *Gaussian → Swiss Roll, Gaussian → Mixture of 8 Gaussians.* We provide qualitative results in Figure 2 and Figure 5 (Appendix E), respectively. In both cases, we provide solutions of the problem for $\epsilon = 0, 0.01, 0.1$ and sample trajectories. For $\epsilon = 0$, all the trajectories are straight lines as they represent solutions for non-regularized OT (1), see [43, §5.4]. For bigger $\epsilon$, trajectories, as expected, become more noisy and less straight.

### 5.2 High-dimensional Gaussians

For general continuous distributions $\mathbb{P}_0, \mathbb{P}_1$, the ground truth solution of entropic OT (2) and DSB (11) is unknown. This makes it challenging to assess how well does our algorithm solve these problems. Fortunately, when $\mathbb{P}_0$ and $\mathbb{P}_1$ are Gaussians, there exist *closed form solutions* of these related problems, see [25] and [9]. Thus, to quantify the performance of our algorithm, we consider entropic OT problems in dimensions $D \in \{2, 16, 64, 128\}$ with $\epsilon = 1$ for Gaussian $\mathbb{P}_0 = \mathcal{N}(0, \Sigma_0)$, $\mathbb{P}_1 = \mathcal{N}(0, \Sigma_1)$. We pick $\Sigma_0, \Sigma_1$ at random: their eigenvectors are uniformly distributed on the unit sphere and eigenvalues are sampled from the loguniform distribution on $[-\log 2, \log 2]$.

| Dim | 2 | 16 | 64 | 128 |
|---|---|---|---|---|
| ENOT (ours) | **0.01** (±0.006) | 0.09 (±0.02) | 0.23 (±0.03) | 0.50 (±0.08) |
| LSOT [45] | 1.82 | 6.42 | 32.18 | 64.32 |
| SCONES [14] | 1.74 | 1.87 | 6.27 | 6.88 |
| MLE-SB [48] | 0.41 | 0.50 | 1.16 | 2.13 |
| DiffSB [15] | 0.7 | 1.11 | 1.98 | 2.20 |
| FB-SDE-A [10] | 0.87 | 0.94 | 1.85 | 1.95 |
| FB-SDE-J [10] | 0.03 | **0.05** | **0.19** | **0.39** |

Table 1: Comparisons of $\text{BW}_2^2$-UVP $\downarrow$ (%) between the target $\mathbb{P}_1$ and learned marginal $\pi_1$.

| Dim | 2 | 16 | 64 | 128 |
|---|---|---|---|---|
| ENOT (ours) | **0.012** (±0.003) | **0.05** (±0.01) | **0.13** (±0.014) | **0.29** (±0.04) |
| LSOT [45] | 6.77 | 14.56 | 25.56 | 47.11 |
| SCONES [14] | 0.92 | 1.36 | 4.62 | 5.33 |
| MLE-SB [48] | 0.3 | 0.9 | 1.34 | 1.8 |
| DiffSB [15] | 0.88 | 1.7 | 2.32 | 2.43 |
| FB-SDE-A [10] | 0.75 | 1.36 | 2.45 | 2.64 |
| FB-SDE-J [10] | 0.07 | 0.22 | 0.34 | 0.58 |

Table 2: Comparisons of $\text{BW}_2^2$-UVP $\downarrow$ (%) between the the EOT plan $\pi^*$ and learned plan $\pi$.

| $t$, time | 0 | 0.2 | 0.4 | 0.6 | 0.8 | 1 |
|---|---|---|---|---|---|---|
| ENOT (ours) | 0 | **0.01** (±0.001) | **0.023** (±0.005) | **0.042** (±0.007) | **0.067** (±0.015) | 0.096 (±0.019) |
| LSOT [45] | 0 | N/A | N/A | N/A | N/A | 6.42 |
| SCONES [14] | 0 | N/A | N/A | N/A | N/A | 6.88 |
| MLE-SB [48] | 0 | 0.10 | 0.23 | 0.30 | 0.36 | 0.50 |
| DiffSB [15] | 0 | 0.19 | 0.48 | 0.68 | 0.91 | 1.11 |
| FB-SDE-A [10] | 0 | 0.17 | 0.45 | 0.61 | 0.77 | 0.94 |
| FB-SDE-J [10] | 0 | 0.18 | 0.32 | 0.31 | 0.17 | **0.05** |

Table 3: Comparisons of $\text{BW}_2^2$-UVP $\downarrow$ (%) between the learned marginal distributions and the ground truth marginal distributions at the intermediate time moments $t = 0, \frac{2}{10}, \ldots, 1$ in dimension $D = 16$.

**Metrics**. We evaluate **(a)** how precisely our algorithm fits the target distribution $\mathbb{P}_1$ on $\mathbb{R}^D$; **(b)** how well it recovers the entropic OT plan $\pi^*$ which is a Gaussian distribution on $\mathbb{R}^D \times \mathbb{R}^D$; **(c)** how accurate are the learned marginal distributions at intermediate times $t = 0, \frac{1}{10}, \ldots, 1$.

In each of the above-mentioned cases, we compute the $\text{BW}_2^2$-UVP [29, §5] between the learned and the ground truth distributions. For two distributions $\widehat{\chi}$ and $\chi$, it is the Wasserstein-2 distance between their Gaussian approximations which is further normalized by the variance of the distribution $\chi$:

$$\text{B}\mathbb{W}_2^2\text{-UVP}(\widehat{\chi}, \chi) = \frac{100\%}{\frac{1}{2}\text{Var}(\chi)} \mathbb{W}_2^2(\mathcal{N}(\mu_{\widehat{\chi}}, \Sigma_{\widehat{\chi}}), \mathcal{N}(\mu_{\chi}, \Sigma_{\chi})). \tag{17}$$

We estimate the metric by using $10^5$ samples.

**Baselines**. We compare our method ENOT with LSOT [45], SCONES [14], MLE-SB [48], DiffSB[15], FB-SDE [10] (two algorithms, A and J). Results are given in Tables 1, 2 and 3. LSOT and SCONES solve EOT without solving SB, hence there are no results in Table 3 for these methods. Our method achieves low $\text{BW}_2^2$-UVP values indicating that it recovers the ground truth process and entropic plan fairly well. The competitive methods have good results in low dimensions, however they perform worse in high dimensions. Importantly, our methods scores the best results in recovering the OT plan (Table 2) and the marginal distributions of the Schrodinger bridge (Table 3). To illustrate the stable convergence of ENOT, we provide the plot of $\text{BW}_2^2$-UVP between the learned plan and the ground truth plan for ENOT during training in Figure 6 (Appendix E).

### 5.3 Colored MNIST

In this section, we test how the entropy parameter $\epsilon$ affects the stochasticity of the learned plan in higher dimensions. For this, we consider the entropic OT problem between colorized MNIST digits of classes "2" ($\mathbb{P}_0$) and "3" ($\mathbb{P}_1$).

**Effect of parameter $\epsilon$.** For $\epsilon = 0, 1, 10$, we learn our Algorithm 1 on the *train* sets of digits "2" and "3". We show the translated *test* images in Figures 3a, 3b and 3c, respectively. When $\epsilon = 0$, there is no diversity in generated "3" samples (Figure 3a), the color remains since the map tried to minimally change the image in the RGB pixel space. When $\epsilon = 1$, some slight diversity in the shape of "3" appears but the color of the input "2" is still roughly preserved (Figure 3c). For higher $\epsilon$, the diversity of generated samples becomes clear (Figure 3c). In particular, the color of "3" starts to slightly deviate from the input "2". That is, increasing the value $\epsilon$ of the entropy term in (2) expectedly leads to bigger stochasticity in the plan. We add the conditional LPIPS variance [24, Table 1] of generated samples for test datasets by ENOT to show how diversity changes for different $\epsilon$ (Table 4). We provide examples of trajectories learned by ENOT in Figure 7 (Appendix F).

**Metrics and baselines.** We compare our method ENOT with SCONES [14], and DiffSB [15] as these are the only methods which the respective authors applied for *data→data* tasks. To evaluate the results, we use the FID metric [23] which is the Bures-Wasserstein (Freschet) distance between the

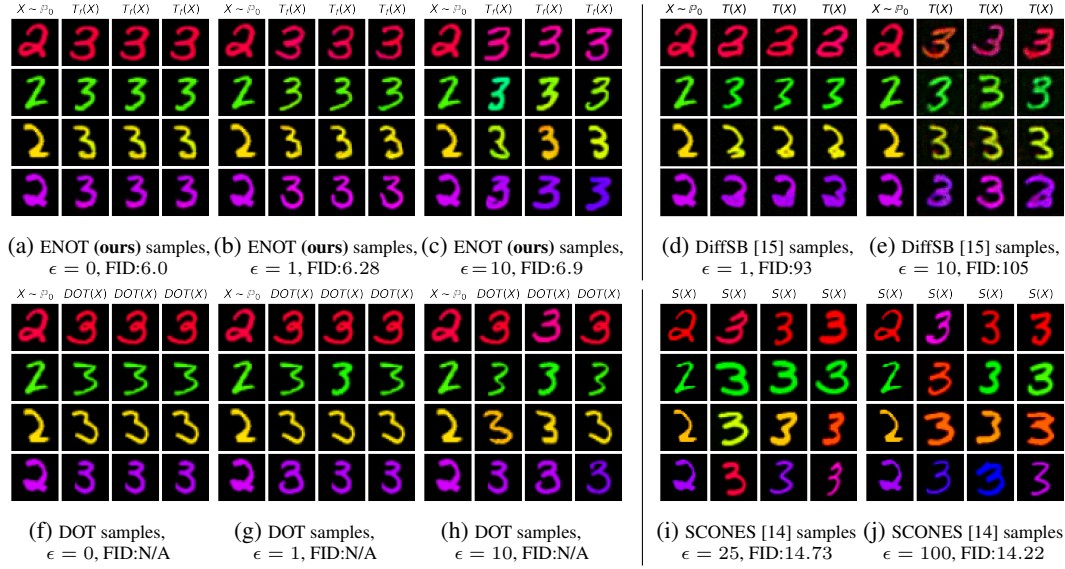

(a) ENOT **(ours)** samples, $\epsilon = 0$, FID:6.0

(b) ENOT **(ours)** samples, $\epsilon = 1$, FID:6.28

(c) ENOT **(ours)** samples, $\epsilon = 10$, FID:6.9

(d) DiffSB [15] samples, $\epsilon = 1$, FID:93

(e) DiffSB [15] samples, $\epsilon = 10$, FID:105

(f) DOT samples, $\epsilon = 0$, FID:N/A

(g) DOT samples, $\epsilon = 1$, FID:N/A

(h) DOT samples, $\epsilon = 10$, FID:N/A

(i) SCONES [14] samples $\epsilon = 25$, FID:14.73

(j) SCONES [14] samples $\epsilon = 100$, FID:14.22

Figure 3: Samples of colored MNIST obtained by ENOT **(ours)** and DOT for different $\epsilon$.

distributions after extracting features using the InceptionV3 model [47]. We measure test FID for every method and present the results and qualitative examples in Figure 3. There are no results for SCONES with $\epsilon = 0, 1, 10$, since it **is not applicable** for such reasonably small $\epsilon$ due to computational instabilities, see [14, §5.1]. DiffSB [15] can be applied for small regularization $\epsilon$, so we test $\epsilon = 1, 10$. By the construction, this algorithm is not suitable for $\epsilon = 0$.

Our ENOT method outperforms the baselines in FID. DiffSB [15] yield very high FID. This is presumably due to instabilities of DiffSB which the authors report in their sequel paper [46]. SCONES yields reasonable quality but due to high $\epsilon = 25, 100$ the shape and color of the generated images "3" starts to deviate from those of their respective inputs "2".

For completeness, we provide the results of the stochastic *matching* of the **test parts** of the datasets by the discrete OT for $\epsilon = 0$ and EOT [12] for $\epsilon = 1, 10$ (Figures 3f, 3g, 3h). This is not the out-of-sample estimation, obtained samples "3" are just **test** samples of "3" (this setup is *unfair*). Discrete OT is *not a competitor* here as it does not generate new samples and uses *target* test samples. Still it gives a rough estimate what to expect from the learned plans for increasing $\epsilon$.

### 5.4 Unpaired Super-resolution of Celeba Faces

For the large-scale evaluation, we adopt the experimental setup of SCONES [14]. We consider the problem of unpaired image super-resolution for the $64 \times 64$ aligned faces of CelebA dataset [35].

| $\epsilon$ | 0 | 1 | 10 |
|---|---|---|---|
| **Colored MNIST** | 0 | $5.3 \cdot 10^{-3}$ | $2.0 \cdot 10^{-2}$ |
| **Celeba** | 0 | $3.4 \cdot 10^{-2}$ | $5.1 \cdot 10^{-2}$ |

Table 4: LPIPS variability of ENOT samples.

We do the *unpaired* train-test split as follows: we split the dataset into 3 parts: 90k (train A1), 90k (train B1), 20k (test C1) samples. For each part we do $2\times$ bilinear downsample and then $2\times$ bilinear upsample to degrade images but keep the original size. As a result, we obtain degraded parts A0, B0, C0. For training in the unpaired setup, we use parts A0 (degraded faces, $\mathbb{P}_0$) and B1 (clean faces, $\mathbb{P}_1$). For testing, we use the hold-out part C0 (unseen samples) with C1 considered as the reference.

We train our model with $\epsilon{=}0, 1, 10$ to and test how it restores C1 (Figure 4a) from C0 images (Figure 4b) and present the qualitative results in Figures 4f, 4g, 4h. We provide examples of trajectories learned by ENOT in Figure 1.

**Metrics and baselines.** To quantify the results, as in [14], we compute the FID score [23] between the sets of mapped C0 images and C1 images (Table 5). The FID of ENOT is better than FID values of the other methods, but increases with $\epsilon$ probably due to the increasing variance of gradients during training . As in §5.1 and §5.3, the diversity of samples increases with $\epsilon$. Our method works with small values of $\epsilon$ and provides **reasonable** amount of diversity in the mapped samples which grows with $\epsilon$ (Table 4). As the baseline among other methods for EOT we consider only SCONES, as it is the only EOT/DSB algorithm which has been applied to *data→data* task at $64 \times 64$ resolution. At the same time, we emphasize that SCONES **is not applicable** for small $\epsilon$ due to instabilities, see [14, §5.1]. This makes it **impractical**, as due to high $\epsilon$, its produces up-scaled images (Figures 4c) are

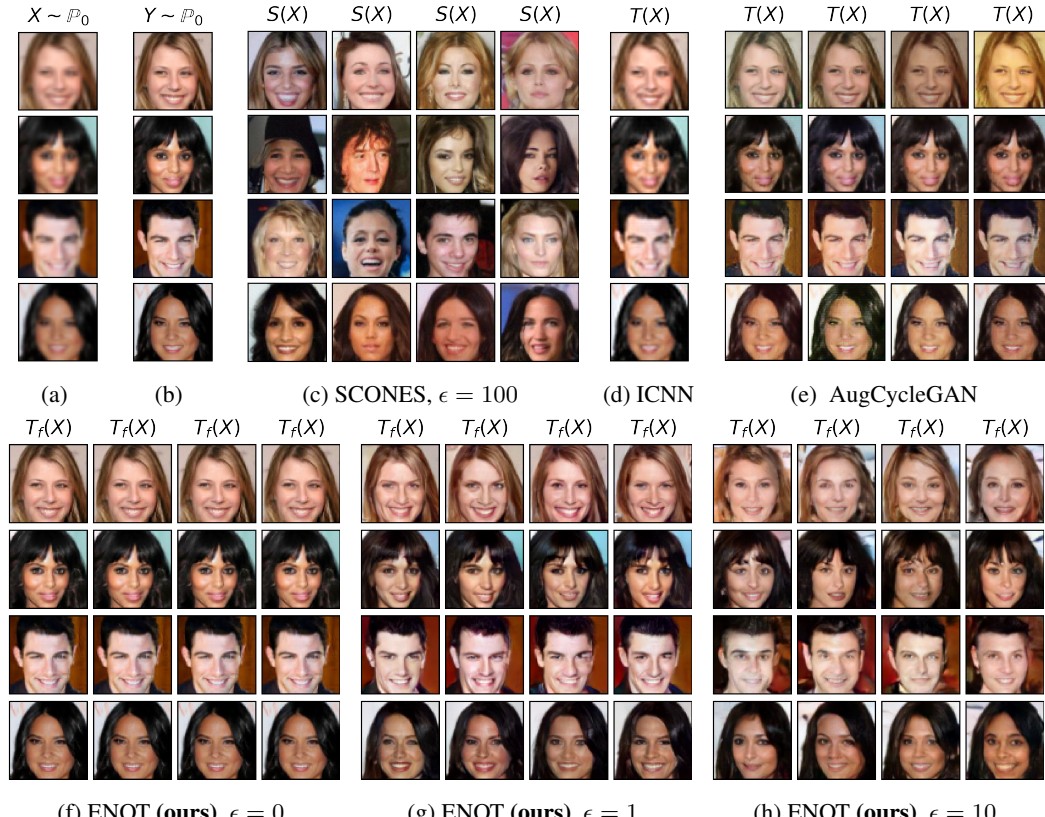

Figure 4: Faces produced by ENOT (**ours**) and SCONEs for various $\epsilon$.
Figure 4a shows test degraded images (C0), 4b – their original high-resolution counterparts (C1).

| Method | ENOT, $\epsilon = 0$ | ENOT, $\epsilon = 1$ | ENOT, $\epsilon = 10$ | SCONES [14], $\epsilon = 100$ | AugCycleGAN [2] | ICNN [38] |
|--------|----------------------|----------------------|-----------------------|-------------------------------|-----------------|-----------|
| FID | **3.78** | **7.63** | **14.8** | 18.88 | 15.2 | 22.2 |

Table 5: Test FID values of various methods in unpaired super-resolution of faces experiment.

nearly random and do not reflect the attributes of the input images (Figure 4a). We do not provide results for DiffSB [15] since it already performs bad on Colored MNIST (§5.3) and the authors also did not consider any image-to-image apart of grayscale 28x28 images.

For completeness, we present results on this setup for other methods, which do not solve EOT: ICNN-based OT [38] and AugCycleGAN [2]. ICNN (4d) learns a deterministic map. AugCycleGAN (4e) learns a stochastic map, but the generated samples differ only by brightness.

## 6 Discussion

**Potential impact.** There is a lack of scalable algorithms for learning continuous entropic OT plans which may be used in *data→data* practical tasks requiring control of the diversity of generated samples. We hope that our results provide a new direction for research towards establishing scalable and efficient methods for entropic OT by using its connection with SB.

**Potential social impact.** Like other popular methods for generating images, our method can be used to simplify the work of designers with digital images and create new products based on it. At the same time, our method may be used for creating fake images just like the other generative models.

**Limitations.** To simulate the trajectories following SDE (9), we use the Euler-Maruyama scheme. It is straightforward but may be imprecise when the number of steps is small or the noise variance $\epsilon$ is high. As a result, for large $\epsilon$, our Algorithm 1 may be computationally heavy due to the necessity to backpropagate through a large computational graph obtained via the simulation. Employing time and memory efficient SDE integration schemes is a promising avenue for the future work.

ACKNOWLEDGEMENTS. This work was partially supported by Skoltech NGP program (Skoltech-MIT joint project).

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

## A  Extended background: KL divergence with the Wiener process plan

This section illustrates that (7) holds. Consider a process $T \in \mathcal{F}(\mathbb{P}_0)$, i.e., $T$ is a probability distribution on $\Omega$ with the marginal $\mathbb{P}_0$ at $t = 0$.

Let $W^\epsilon$ be the Wiener process with variance $\epsilon$ starting at $\mathbb{P}_0$, i.e., it satisfies $dX_t = \sqrt{\epsilon}dW_t$ with $X_0 \sim \mathbb{P}_0$. Hence, $\pi^{W^\epsilon}(y|x)$ is the normal distribution $\frac{d\pi^{W^\epsilon}(y|x)}{dy} = \mathcal{N}(y|x, \epsilon I)$. Then $\mathrm{KL}(\pi^T || \pi^{W^\epsilon})$ between joint distributions at times $t = 0$ and $t = 1$ of these processes is given by:

$$\mathrm{KL}(\pi^T || \pi^{W^\epsilon}) = -\int_{\mathcal{X} \times \mathcal{Y}} \log \frac{d\pi^{W^\epsilon}(x,y)}{d[x,y]} d\pi^T(x,y) + \underbrace{\int_{\mathcal{X} \times \mathcal{Y}} \log \frac{d\pi^T(x,y)}{d[x,y]} d\pi^T(x,y)}_{=-H(\pi^T)}, \quad (18)$$

where $\frac{d\pi(x,y)}{d[x,y]}$ denotes the joint density of distribution $\pi$. We derive

$$-\int_{\mathcal{X} \times \mathcal{Y}} \log \frac{d\pi^{W^\epsilon}(x,y)}{d[x,y]} d\pi^T(x,y) = -\int_{\mathcal{X} \times \mathcal{Y}} \log \frac{d\pi^{W^\epsilon}(y|x)}{dy} \frac{d\pi^{W^\epsilon}(x)}{dx} d\pi^T(x,y) =$$

$$-\int_{\mathcal{X} \times \mathcal{Y}} \log \frac{d\pi^{W^\epsilon}(y|x)}{dy} d\pi^T(x,y) - \int_{\mathcal{X}} \int_{\mathcal{Y}} \log \frac{d\pi^{W^\epsilon}(x)}{dx} d\pi^T(y|x) \overbrace{d\mathbb{P}_0(x)}^{d\pi_0^T(x)} =$$

$$-\int_{\mathcal{X} \times \mathcal{Y}} \log \frac{d\pi^{W^\epsilon}(y|x)}{dy} d\pi^T(x,y) - \int_{\mathcal{X}} \log \frac{d\pi^{W^\epsilon}(x)}{dx} \Big[ \int_{\mathcal{Y}} 1 d\pi^T(y|x) \Big] d\mathbb{P}_0(x) =$$

$$-\int_{\mathcal{X}} \int_{\mathcal{Y}} \log \frac{d\pi^{W^\epsilon}(y|x)}{dy} d\pi^T(x,y) - \int_{\mathcal{X}} \log \frac{d\pi^{W^\epsilon}(x)}{dx} d\mathbb{P}_0(x) =$$

$$-\int_{\mathcal{X}} \int_{\mathcal{Y}} \log \frac{d\pi^{W^\epsilon}(y|x)}{dy} d\pi^T(x,y) - \int_{\mathcal{X}} \log \frac{d\mathbb{P}_0(x)}{dx} d\mathbb{P}_0(x) =$$

$$-\int_{\mathcal{X} \times \mathcal{Y}} \log \frac{d\pi^{W^\epsilon}(y|x)}{dy} d\pi^T(x,y) + H(\mathbb{P}_0) =$$

$$-\int_{\mathcal{X} \times \mathcal{Y}} \log \Big( (2\pi\epsilon)^{\frac{-D}{2}} \exp \Big( -\frac{||x-y||^2}{2\epsilon} \Big) \Big) d\pi^T(x,y) + H(\mathbb{P}_0) =$$

$$+\frac{D}{2} \log(2\pi\epsilon) + \int_{\mathcal{X} \times \mathcal{Y}} \frac{||x-y||^2}{2\epsilon} d\pi^T(x,y) + H(\mathbb{P}_0).$$

After substituting this result into (18), one obtains:

$$\mathrm{KL}(\pi^T || \pi^{W^\epsilon}) = \int_{\mathcal{X} \times \mathcal{Y}} \frac{||x-y||^2}{2\epsilon} d\pi^T(x,y) - H(\pi^T) + \underbrace{\frac{D}{2} \log(2\pi\epsilon) + H(\mathbb{P}_0)}_{=C \text{ in (7)}}. \quad (19)$$

## B  Proofs

In this section, we provide the proof for our main theoretical results (Theorems 4.1 and 4.3). The proofs require several auxiliary results which we formulate and prove in §B.1 and §B.2.

In §B.1, we show that entropic OT can be reformulated as a maximin problem. This is a technical intermediate result needed to derive our main maximin reformulation of SB (Theorem 4.1). More precisely, in §B.2, we show that these maximin problems for entropic OT and SB are actually equivalent. By using this observation and related facts, in §B.3, we prove our Theorems 4.1 and 4.3.

### B.1  Relaxation of entropic OT

To begin with, we recall some facts regarding EOT and SB. Recall the definition of EOT (2):

$$\inf_{\pi \in \Pi(\mathbb{P}_0, \mathbb{P}_1)} \int_{\mathcal{X} \times \mathcal{Y}} \frac{||x-y||^2}{2} d\pi(x,y) - \epsilon H(\pi). \quad (20)$$

Henceforth, we assume that $\mathbb{P}_0$ and $\mathbb{P}_1$ are *absolutely continuous*. The situation when $\mathbb{P}_0$ or $\mathbb{P}_1$ is not absolutely continuous is not of any practical interest: there is no $\pi \in \Pi(\mathbb{P}_0, \mathbb{P}_1)$ for which the differential entropy $H(\pi)$ is finite which means that (20) equals to $+\infty$ for every $\pi \in \Pi(\mathbb{P}_0, \mathbb{P}_1)$, i.e., every plan is optimal. In turn, when $\mathbb{P}_0$ and $\mathbb{P}_1$ are absolutely continuous, the OT plan is unique thanks to the strict convexity of entropy (on the set of absolutely continuous plans).

Recall equation (19) for $\mathrm{KL}(\pi||\pi^{W^\epsilon})$:

$$\mathrm{KL}(\pi||\pi^{W^\epsilon}) = \int_{\mathcal{X} \times \mathcal{Y}} \frac{||x - y||^2}{2\epsilon} d\pi(x, y) - H(\pi) + C. \tag{21}$$

We again note that

$$\inf_{\pi \in \Pi(\mathbb{P}_0, \mathbb{P}_1)} \mathrm{KL}(\pi||\pi^{W^\epsilon}) = \frac{1}{\epsilon} \inf_{\pi \in \Pi(\mathbb{P}_0, \mathbb{P}_1)} \left\{ \int_{\mathcal{X} \times \mathcal{Y}} \frac{||x - y||^2}{2} d\pi(x, y) - \epsilon H(\pi) \right\} + C,$$

i.e., problems (20) and (21) can be viewed as equivalent as their minimizers are the same. For convenience, we proceed with $\inf_{\pi \in \Pi(\mathbb{P}_0, \mathbb{P}_1)} \mathrm{KL}(\pi||\pi^{W^\epsilon})$ and denote its optimal value by $\mathcal{L}^*$, i.e.,

$$\mathcal{L}^* \stackrel{\text{def}}{=} \inf_{\pi \in \Pi(\mathbb{P}_0, \mathbb{P}_1)} \mathrm{KL}(\pi||\pi^{W^\epsilon}).$$

For a given $\beta \in \mathcal{C}_{b,2}(\mathcal{Y})$, we define an auxiliary joint distribution $d\pi^\beta(x, y) = d\pi^\beta(y|x) d\mathbb{P}_0(x)$, where $d\pi^\beta(y|x)$ is given by

$$d\pi^\beta(y|x) = \frac{1}{C_\beta^x} \exp(\beta(y)) d\pi^{W^\epsilon}(y|x),$$

where $C_\beta^x(x) \stackrel{\text{def}}{=} \int_{\mathcal{Y}} \exp(\beta(y)) d\pi^{W^\epsilon}(y|x)$. Note that $C_\beta^x < \infty$ since $\beta \in \mathcal{C}_{b,2}(\mathcal{Y})$ is upper bounded.

Before going further, we need to introduce several technical auxiliary results.

**Proposition B.1.** *For $\nu \in \mathcal{P}_2(\mathcal{Y})$ and $x \in \mathcal{X}$ it holds that*

$$\mathrm{KL}(\nu||\pi^{W^\epsilon}(\cdot|x)) - \int_{\mathcal{Y}} \beta(y) d\nu(y) = \mathrm{KL}(\nu||\pi^\beta(\cdot|x)) - \log C_\beta^x. \tag{22}$$

*Proof of Proposition B.1.* We derive

$$\mathrm{KL}(\nu||\pi^{W^\epsilon}(\cdot|x)) - \int_{\mathcal{Y}} \beta(y) d\nu(y) = \int_{\mathcal{Y}} \log \frac{d\nu(y)}{d\pi^{W^\epsilon}(y|x)} d\nu(y) - \int_{\mathcal{Y}} \beta(y) d\nu(y) =$$

$$\int_{\mathcal{Y}} \log \frac{d\nu(y)}{\exp(\beta(y)) d\pi^{W^\epsilon}(y|x)} d\nu(y) = \int_{\mathcal{Y}} \log \frac{C_\beta^x d\nu(y)}{C_\beta^x \exp(\beta(y)) d\pi^{W^\epsilon}(y|x)} d\nu(y) =$$

$$\int_{\mathcal{Y}} \log \frac{d\nu(y)}{d\pi^\beta(y|x)} d\nu(y) - \log C_\beta^x = \mathrm{KL}(\nu||\pi^\beta(\cdot|x)) - \log C_\beta^x.$$

$\square$

**Lemma B.2.** *For $\pi \in \Pi(\mathbb{P}_0)$, i.e., probability distributions $\pi \in \mathcal{P}_2(\mathcal{X} \times \mathcal{Y})$ whose projection to $\mathcal{X}$ equals $\mathbb{P}_0$, we have*

$$\mathrm{KL}(\pi||\pi^{W^\epsilon}) - \int_{\mathcal{Y}} \beta(y) d\pi(y) = \mathrm{KL}(\pi||\pi^\beta) - \int_{\mathcal{X}} \log C_\beta^x d\mathbb{P}_0(x). \tag{23}$$

*Proof of Lemma B.2.* For each $x \in \mathcal{X}$, we substitute $\nu = \pi(\cdot|x)$ to (22) and integrate over $x \sim \mathbb{P}_0$. For the left part, we obtain the following:

$$\int_{\mathcal{X}} \left( \mathrm{KL}(\pi(\cdot|x)||\pi^{W^\epsilon}(\cdot|x)) - \int_{\mathcal{Y}} \beta(y) d\pi(y|x) \right) d\mathbb{P}_0(x) =$$

$$\int_{\mathcal{X}} \mathrm{KL}(\pi(\cdot|x)||\pi^{W^\epsilon}(\cdot|x)) d\mathbb{P}_0(x) - \int_{\mathcal{X} \times \mathcal{Y}} \beta(y) d\pi(y|x) d\mathbb{P}_0(x) =$$

$$\int_{\mathcal{X}} \int_{\mathcal{Y}} \log \frac{d\pi(y|x)}{d\pi^{W^\epsilon}(y|x)} d\pi(y|x) d\mathbb{P}_0(x) - \int_{\mathcal{Y}} \beta(y) d\pi_1(y) =$$

$$\int_{\mathcal{X}} \int_{\mathcal{Y}} \log \frac{d\pi(y|x) d\mathbb{P}_0(x)}{d\pi^{W^\epsilon}(y|x) d\mathbb{P}_0(x)} d\pi(y|x) d\mathbb{P}_0(x) - \int_{\mathcal{Y}} \beta(y) d\pi_1(y) =$$

$$\int_{\mathcal{X} \times \mathcal{Y}} \log \frac{d\pi(x, y)}{d\pi^{W^\epsilon}(x, y)} d\pi(x, y) - \int_{\mathcal{Y}} \beta(y) d\pi_1(y) =$$

$$\mathrm{KL}(\pi || \pi^{W^\epsilon}) - \int_{\mathcal{Y}} \beta(y) d\pi_1(y).$$

For the right part, we obtain:

$$\int_{\mathcal{X}} \left\{ \mathrm{KL}(\pi(\cdot|x) || \pi^\beta(\cdot|x)) - \log C_\beta^x \right\} d\mathbb{P}_0(x) =$$

$$\int_{\mathcal{X}} \mathrm{KL}(\pi(\cdot|x) || \pi^\beta(\cdot|x)) d\mathbb{P}_0(x) - \int_{\mathcal{X}} \log C_\beta^x d\mathbb{P}_0(x) =$$

$$\int_{\mathcal{X}} \int_{\mathcal{Y}} \log \frac{d\pi(y|x)}{d\pi^\beta(y|x)} d\pi(y|x) d\mathbb{P}_0(x) - \int_{\mathcal{X}} \log C_\beta^x d\mathbb{P}_0(x) =$$

$$\int_{\mathcal{X}} \int_{\mathcal{Y}} \log \frac{d\pi(y|x) d\mathbb{P}_0(x)}{d\pi^\beta(y|x) d\mathbb{P}_0(x)} d\pi(y|x) d\mathbb{P}_0(x) - \int_{\mathcal{X}} \log C_\beta^x d\mathbb{P}_0(x) =$$

$$\int_{\mathcal{X} \times \mathcal{Y}} \log \frac{d\pi(x, y)}{d\pi^\beta(x, y)} d\pi(x, y) - \int_{\mathcal{X}} \log C_\beta^x d\mathbb{P}_0(x) =$$

$$\mathrm{KL}(\pi || \pi^\beta) - \int_{\mathcal{X}} \log C_\beta^x d\mathbb{P}_0(x).$$

Hence, the equality (23) holds. $\qquad \square$

Now we introduce the following auxiliary functional $\widetilde{\mathcal{L}}$:

$$\widetilde{\mathcal{L}}(\beta, \pi) \stackrel{\mathrm{def}}{=} \mathrm{KL}(\pi || \pi^{W^\epsilon}) - \int_{\mathcal{Y}} \beta(y) d\pi_1(y) + \int_{\mathcal{Y}} \beta(y) d\mathbb{P}_1(y).$$

Recall that $\pi_1$ denotes the second marginal distribution of $\pi$. We use this functional to derive the saddle point reformulation of EOT.

**Lemma B.3** (Relaxation of entropic optimal transport). *It holds that*

$$\mathcal{L}^* = \inf_{\pi \in \Pi(\mathbb{P}_0, \mathbb{P}_1)} \mathrm{KL}(\pi || \pi^{W^\epsilon}) = \sup_\beta \inf_{\pi \in \Pi(\mathbb{P}_0)} \widetilde{\mathcal{L}}(\beta, \pi), \tag{24}$$

*where* $\sup$ *is taken over potentials* $\beta \in \mathcal{C}_{b,2}(\mathcal{Y})$ *and* $\inf$ *over* $\pi \in \Pi(\mathbb{P}_0)$.

*Proof of Lemma B.3.* We obtain

$$\inf_{\pi \in \Pi(\mathbb{P}_0, \mathbb{P}_1)} \mathrm{KL}(\pi || \pi^{W^\epsilon}) = \inf_{\pi \in \Pi(\mathbb{P}_0, \mathbb{P}_1)} \left\{ \int_{\mathcal{X}} \mathrm{KL}(\pi(y|x) || \pi^{W^\epsilon}(y|x)) d\mathbb{P}_0(x) \right\} =$$

$$\inf_{\pi \in \Pi(\mathbb{P}_0, \mathbb{P}_1)} \int_{\mathcal{X}} C(x, \pi(y|x)) d\mathbb{P}_0(x), \tag{25}$$

where $C(x, \nu) \stackrel{\mathrm{def}}{=} \mathrm{KL}(\nu || \pi^{W^\epsilon}(y|x))$. The last problem in (25) is known as weak OT [7, 20] with a weak OT cost $C$. For a given $\beta \in \mathcal{C}_{b,2}(\mathcal{Y})$, consider its weak $C$-transform given by:

$$\beta^C(x) \stackrel{\mathrm{def}}{=} \inf_{\nu \in \mathcal{P}_2(\mathcal{Y})} \left\{ C(x, \nu) - \int_{\mathcal{Y}} \beta(y) d\nu(y) \right\}. \tag{26}$$

Since $C : \mathcal{X} \times \mathcal{P}_2(\mathcal{Y}) \to \mathbb{R}$ is lower bounded (by zero), convex in the second argument and jointly lower semi-continuous, the following equality holds [7, Theorem 1.3]:

$$\mathcal{L}^* = \inf_{\pi \in \Pi(\mathbb{P}_0, \mathbb{P}_1)} \int_{\mathcal{X}} C(x, \pi(y|x)) d\mathbb{P}_0(x) = \sup_\beta \left\{ \int_{\mathcal{X}} \beta^C(x) d\mathbb{P}_0(x) + \int_{\mathcal{Y}} \beta(y) d\mathbb{P}_1(y) \right\}, \tag{27}$$

where $\sup$ is taken over $\beta \in \mathcal{C}_{b,2}(\mathcal{Y})$. We use our Proposition B.1 to note that

$$\beta^C(x) = \inf_{\nu \in \mathcal{P}_2(\mathcal{Y})} \left\{ \mathrm{KL}\big(\nu || \pi^{W^\epsilon}(y|x)\big) - \int_{\mathcal{Y}} \beta(y) d\nu(y) \right\} =$$
$$\inf_{\nu \in \mathcal{P}_2(\mathcal{Y})} \left\{ \mathrm{KL}(\nu || \pi^\beta(\cdot|x)) - \log C_\beta^x \right\} = -\log C_\beta^x.$$

This allows us to derive

$$\int_{\mathcal{X}} \beta^C(x) d\mathbb{P}_0(x) + \int_{\mathcal{Y}} \beta(y) d\mathbb{P}_1(y) = \qquad (28)$$
$$-\int_{\mathcal{X}} \log C_\beta^x d\mathbb{P}_0(x) + \int_{\mathcal{Y}} \beta(y) d\mathbb{P}_1(y) =$$
$$\overbrace{\left\{ \inf_{\pi \in \Pi(\mathbb{P}_0)} \mathrm{KL}(\pi || \pi^\beta) \right\}}^{=0} - \int_{\mathcal{X}} \log C_\beta^x d\mathbb{P}_0(x) + \int_{\mathcal{Y}} \beta(y) d\mathbb{P}_1(y) =$$
$$\inf_{\pi \in \Pi(\mathbb{P}_0)} \left\{ \mathrm{KL}(\pi || \pi^\beta) - \overbrace{\int_{\mathcal{X}} \log C_\beta^x d\mathbb{P}_0(x) + \int_{\mathcal{Y}} \beta(y) d\mathbb{P}_1(y)}^{\text{Do not depend on } \pi} \right\} =$$
$$\inf_{\pi \in \Pi(\mathbb{P}_0)} \left\{ \mathrm{KL}(\pi || \pi^{W^\epsilon}) - \int_{\mathcal{Y}} \beta(y) d\pi(y) + \int_{\mathcal{Y}} \beta(y) d\mathbb{P}_1(y) \right\} = \inf_{\pi \in \Pi(\mathbb{P}_0)} \widetilde{\mathcal{L}}(\beta, \pi). \qquad (29)$$

Here in transition to line (29), we use our Lemma B.2. It remains to take $\sup_\beta$ in equality between (28) and (29) and then recall (27) to finish the proof and obtain desired (24). $\qquad\square$

Thus, we can obtain the value $\mathcal{L}^*$ (8) by solving maximin problem (24) with only one constraint $\pi \in \Pi(\mathbb{P}_0)$. Moreover, our following lemma shows that in all optimal pairs $(\beta^*, \pi^*)$ which solve maximin problem (24), $\pi^*$ is necessary the unique entropic OT plan between $\mathbb{P}_0$ and $\mathbb{P}_1$.

**Lemma B.4** (Entropic OT plan solves the relaxed entropic OT problem)**.** *Let $\pi^*$ be the entropic OT plan between $\mathbb{P}_0$ and $\mathbb{P}_1$. For every optimal $\beta^* \in \mathrm{argsup}_\beta \inf_{\pi \in \Pi(\mathbb{P}_0)} \mathcal{L}(\beta, \pi)$, we have*

$$\pi^* = \mathrm{arginf}_{\pi \in \Pi(\mathbb{P}_0)} \widetilde{\mathcal{L}}(\beta^*, \pi). \qquad (30)$$

*Proof of Lemma B.4.* Since $\beta^*$ is optimal, we know from Lemma B.3 that $\inf_{\pi \in \Pi(\mathbb{P}_0)} \mathcal{L}(\beta^*, \pi) = \mathcal{L}^*$. Thanks to $\pi^* \in \Pi(\mathbb{P}_0, \mathbb{P}_1)$, we have $\pi_1^* = \mathbb{P}_1$. We substitute $\pi^*$ to $\mathcal{L}(\beta^*, \pi)$ and obtain

$$\mathcal{L}(\beta^*, \pi^*) = \mathrm{KL}(\pi^* || \pi^{W^\epsilon}) + \int_{\mathcal{Y}} \beta(y) d\mathbb{P}_1(y) - \int_{\mathcal{Y}} \beta(y) \overbrace{d\pi_1^*(y)}^{=d\mathbb{P}_1(y)} = \mathrm{KL}(\pi^* || \pi^{W^\epsilon}) = \mathcal{L}^*. \qquad (31)$$

The functional $\pi \mapsto \mathcal{L}(\beta^*, \pi)$ is strictly convex (in the convex subset of $\Pi(\mathbb{P}_0)$ of distributions $\pi$ for which $\mathrm{KL}(\pi || \pi^{W^\epsilon}) < \infty$). Thus, it has a unique minimizer, which is $\pi^*$. $\qquad\square$

From our Lemmas B.3 and B.4 it follows that to get the OT plan $\pi^*$ one may solve the maximin problem (24) to obtain an optimal saddle point $(\beta^*, \pi^*)$. Unfortunately, it is challenging to estimate $\mathrm{KL}(\pi || \pi^{W^\epsilon})$ from samples, which limits the usage of this objective in practice.

## B.2 Equivalence of EOT and DSB relaxed problems

Below we show how to relax SB problem (11) and link its solution to the relaxed entropic OT (24).

For a given $\beta \in \mathcal{C}_{b,2}(\mathcal{Y})$, we define an auxiliary process $T^\beta$ such that its conditional distributions are $T_{|x,y}^\beta = W_{|x,y}^\epsilon$ and its joint distribution $\pi^{T^\beta}$ at $t = 0, 1$ is given by $\pi^\beta$.

To simplify many of upcoming formulas, we introduce $C_\beta \stackrel{\text{def}}{=} \int_{\mathcal{X}} \log C_\beta^x d\mathbb{P}_0(x)$. Also, we introduce $\mathcal{F}(\mathbb{P}_0)$ to denote the set of processes starting at $\mathbb{P}_0$ at time $t = 0$.

**Lemma B.5** (Inner objectives of relaxed EOT and SB are KL with $T^\beta$ and $\pi^{T^\beta}$). *For $\pi \in \Pi(\mathbb{P}_0)$ and $T \in \mathcal{F}(\mathbb{P}_0)$, the following equations hold:*

$$\widetilde{\mathcal{L}}(\beta, \pi) = \text{KL}(\pi || \pi^{T^\beta}) - C_\beta + \int_{\mathcal{Y}} \beta(y) d\mathbb{P}_1(y), \tag{32}$$

$$\mathcal{L}(\beta, T) = \text{KL}(T || T^\beta) - C_\beta + \int_{\mathcal{Y}} \beta(y) d\mathbb{P}_1(y). \tag{33}$$

*Note that the last two terms in each line depend only on $\beta$ but not on $\pi$ or $T$.*

*Proof of Lemma B.5.* The first equation (32) directly follows from Lemma B.2. Now we prove (33):

$$\mathcal{L}(\beta, T) - \int_{\mathcal{Y}} \beta(y) d\mathbb{P}_1(y) = \text{KL}(T || W^\epsilon) - \int \beta(y) d\pi_1^T(y) =$$

$$\text{KL}(\pi^T || \pi^{W^\epsilon}) + \int_{\mathcal{X} \times \mathcal{Y}} \text{KL}(T_{|x,y} || W_{|x,y}^\epsilon) d\pi^T(x,y) - \int \beta(y) d\pi_1^T(y) = \tag{34}$$

$$\text{KL}(\pi^T || \pi^{T^\beta}) - C_\beta + \int_{\mathcal{X} \times \mathcal{Y}} \text{KL}(T_{|x,y} || W_{|x,y}^\epsilon) d\pi^T(x,y) =$$

$$\text{KL}(\pi^T || \pi^{T^\beta}) - C_\beta + \int_{\mathcal{X} \times \mathcal{Y}} \text{KL}(T_{|x,y} || T_{|x,y}^\beta) d\pi^T(x,y) = \text{KL}(T || T^\beta) - C_\beta. \tag{35}$$

In the transition to line (34), we use the disintegration formula (6). In line (35), we use the definition of $T^\beta$, i.e., we exploit the fact that $T_{|x,y}^\beta = W_{|x,y}^\epsilon$ and again use (6). $\qquad \square$

As a result of Lemma B.5, we obtain the following important corollary.

**Corollary B.6** (The solution to the inner problem of relaxed SB is a diffusion). *Consider the problem*

$$\inf_{T \in \mathcal{F}(\mathbb{P}_0)} \mathcal{L}(\beta, T). \tag{36}$$

*Then $T^\beta$ is the unique optimizer of (36) and it holds that $T^\beta \in \mathcal{D}(\mathbb{P}_0)$, i.e., it is a diffusion process:*

$$T^\beta = \underset{T \in \mathcal{F}(\mathbb{P}_0)}{\text{arginf}} \, \mathcal{L}(\beta, T) = \underset{T_f \in \mathcal{D}(\mathbb{P}_0)}{\text{arginf}} \, \mathcal{L}(\beta, T_f). \tag{37}$$

*Proof.* Thanks to (33), we see that $T^\beta$ is the unique minimizer of (36). Now let $\mathbb{Q} \overset{\text{def}}{=} \pi_1^{T^\beta}$. Then

$$T^\beta = \underset{T \in \mathcal{F}(\mathbb{P}_0)}{\text{arginf}} \, \mathcal{L}(\beta, T) = \underset{T \in \mathcal{F}(\mathbb{P}_0)}{\text{arginf}} \left[ \text{KL}(T || W^\epsilon) - \int_{\mathcal{Y}} \beta(y) d\pi_1^T(y) \right] =$$

$$\underset{T \in \mathcal{F}(\mathbb{P}_0, \mathbb{Q})}{\text{arginf}} \left[ \text{KL}(T || W^\epsilon) - \underbrace{\int_{\mathcal{Y}} \beta(y) d\pi_1^T(y)}_{=\text{Const, since } \pi_1^T = \pi_1^{T^\beta} = \mathbb{Q}} \right] = \underset{T \in \mathcal{F}(\mathbb{P}_0, \mathbb{Q})}{\text{arginf}} \, \text{KL}(T || W^\epsilon) =$$

$$\underset{T_f \in \mathcal{D}(\mathbb{P}_0, \mathbb{Q})}{\text{arginf}} \, \text{KL}(T_f || W^\epsilon) = \underset{T_f \in \mathcal{D}(\mathbb{P}_0, \mathbb{Q})}{\text{arginf}} \, \frac{1}{2\epsilon} \mathbb{E}_{T_f} \Big[ \int_0^1 ||f(X_t, t)||^2 dt \Big]. \tag{38}$$

In transition to (38), we use the fact that the process solving the Schrödinger Bridge (this time between $\mathbb{P}_0$ and $\mathbb{Q}$) with the Wiener Prior is a diffusion process (see Dynamic SB problem in §2.2 for details). As a result, we obtain $T^\beta \in \mathcal{D}(\mathbb{P}_0, \mathbb{Q}) \subset \mathcal{D}(\mathbb{P}_0)$ and finish the proof. $\qquad \square$

Below we show that for a given $\beta$, minimization of the SB relaxed functional $\mathcal{L}(\beta, T_f)$ over $T_f$ is equivalent to the minimization of relaxed EOT functional $\widetilde{\mathcal{L}}(\beta, \pi)$ (24) with the same $\beta$.

**Lemma B.7** (Equivalence of the inf values of the relaxed functionals). *It holds that*

$$\inf_{T_f \in \mathcal{D}(\mathbb{P}_0)} \mathcal{L}(\beta, T_f) = \inf_{\pi \in \Pi(\mathbb{P}_0)} \widetilde{\mathcal{L}}(\beta, \pi) = -C_\beta + \int \beta(y) d\mathbb{P}_1(y). \tag{39}$$

*Moreover, the unique minimizers are given by $T^\beta \in \mathcal{D}(\mathbb{P}_0)$ and $\pi^{T^\beta} \in \Pi(\mathbb{P}_0)$, respectively.*

*Proof of Lemma B.7.* Follows from Lemma B.5 and Corollary B.6. □

Finally, we see that both the maximin problems are equivalent.

**Corollary B.8** (Equivalence of EOT and DSB maximin problems)**.** *It holds that*

$$\mathcal{L}^* = \sup_\beta \inf_{T_f \in \mathcal{D}(\mathbb{P}_0)} \mathcal{L}(\beta, T_f) = \sup_\beta \inf_{\pi \in \Pi(\mathbb{P}_0)} \widetilde{\mathcal{L}}(\beta, \pi) \tag{40}$$

*Proof of Corollary B.8.* We take $\sup_\beta$ of both parts in equation (39). □

Also, it follows that the maximization of $\inf_{T_f \in \mathcal{D}(\mathbb{P}_0)} \mathcal{L}(\beta, T_f)$ over $\beta$ allows to solve entropic OT.

## B.3 Proofs of main results

Finally, after long preparations, we prove our main Theorem 4.1.

*Proof of Theorem 4.1 and Corollary 4.2.* From our Lemma B.7 and Corollary B.8 it follows that

$$\beta^* \in \operatorname*{argsup}_\beta \inf_{T_f \in \mathcal{D}(\mathbb{P}_0)} \mathcal{L}(\beta, T_f) \Leftrightarrow \beta^* \in \operatorname*{argsup}_\beta \inf_{\pi \in \Pi(\mathbb{P}_0)} \widetilde{\mathcal{L}}(\beta, \pi),$$

i.e., both maximin problems share the same optimal $\beta^*$. Thanks to our Lemma B.7, we already know that the process $T^{\beta^*} \in \mathcal{D}(\mathbb{P}_0)$ and the plan $\pi^{T^{\beta^*}} \in \Pi(\mathbb{P}_0)$ are the unique minimizers of problems

$$\inf_{T_f \in \mathcal{D}(\mathbb{P}_0)} \mathcal{L}(\beta^*, T_f) = \inf_{\pi \in \Pi(\mathbb{P}_0)} \widetilde{\mathcal{L}}(\beta^*, \pi),$$

respectively. Therefore, $T_{f^*} = T^{\beta^*}$ and, in particular, $\pi^{T_{f^*}} = \pi^{T^{\beta^*}}$. Moreover, since $(\beta^*, \pi^{T_{f^*}})$ is an optimal saddle point for $\widetilde{\mathcal{L}}$, from Lemma B.4 we conclude that $\pi^{T_{f^*}} = \pi^*$, i.e., $\pi^{T_{f^*}}$ is the **EOT plan** between $\mathbb{P}_0$ and $\mathbb{P}_1$. In particular, $\pi^{T_{f^*}} \in \Pi(\mathbb{P}_0, \mathbb{P}_1)$ which also implies that $T_{f^*} \in \mathcal{D}(\mathbb{P}_0, \mathbb{P}_1)$. The last step is to derive

$$\mathcal{L}^* = \mathcal{L}(\beta^*, T_{f^*}) = \mathrm{KL}(T_{f^*} || W^\epsilon) + \underbrace{\int_{\mathcal{Y}} \beta^*(y) d\mathbb{P}_1(y) - \int_{\mathcal{Y}} \beta^*(y) \overbrace{d\pi_1^{T_{f^*}}(y)}^{=d\mathbb{P}_1(y)}}_{=0 \text{ since } T_{f^*} \in \mathcal{D}(\mathbb{P}_0, \mathbb{P}_1)} = \mathrm{KL}(T_{f^*} || W^\epsilon).$$

which concludes that $T_{f^*}$ is the **solution to SB** (5). □

*Proof of Theorem 4.3.* Part 1. From Lemma B.5 and Corollary B.6 it follows that that $\inf_{T_f} \mathcal{L}(\hat{\beta}, T_f)$ has the unique minimizer $T^{\widehat{\beta}}$ whose conditional distributions are $T^{\widehat{\beta}}_{|x,y} = W^\epsilon_{|x,y}$. Therefore,

$$\epsilon_1 = \mathcal{L}(\hat{\beta}, T_{\hat{f}}) - \inf_{T_f} \mathcal{L}(\hat{\beta}, T_f) =$$

$$\left[ \mathrm{KL}(T_{\hat{f}} || T^{\widehat{\beta}}) - C_{\hat{\beta}} + \int_{\mathcal{Y}} \widehat{\beta}(y) d\mathbb{P}_1(y) \right] - \left[ -C_{\widehat{\beta}} + \int_{\mathcal{Y}} \widehat{\beta}(y) d\mathbb{P}_1(y) \right] = \mathrm{KL}(T_{\hat{f}} || T^{\widehat{\beta}}). \tag{41}$$

Part 2. Now we consider $\epsilon_2$. We know that

$$\mathcal{L}^* = \mathrm{KL}(T_{f^*} || W^\epsilon) =$$

$$\mathrm{KL}(\pi^{T_{f^*}} || \pi^{W^\epsilon}) + \int_{\mathcal{X} \times \mathcal{Y}} \mathrm{KL}(T_{f^*|x,y} || W^\epsilon_{|x,y}) d\pi^{T_{f^*}}(x, y) = \mathrm{KL}(\pi^{T_{f^*}} || \pi^{W^\epsilon}).$$

From Lemma B.5 and Corollary B.6, we also know that

$$\inf_{T_f} \mathcal{L}(\hat{\beta}, T_f) = -C_{\hat{\beta}} + \int \hat{\beta}(y) d\mathbb{P}_1(y).$$

Therefore:

$$\epsilon_2 = \mathcal{L}^* - \inf_{T_f} \mathcal{L}(\hat{\beta}, T_{f^*}) = \mathrm{KL}(\pi^{T_{f^*}} || \pi^{W^\epsilon}) + C_{\hat{\beta}} - \int \hat{\beta}(y) d\mathbb{P}_1(y) =$$

$$\text{KL}(\pi^{T_{f^*}}||\pi^{W_\epsilon}) + \int_{\mathcal{X}} \log C_{\hat{\beta}}^x d\mathbb{P}_0(x) - \int \hat{\beta}(y)d\mathbb{P}_1(y) =$$

$$\int_{\mathcal{X}} \text{KL}(\pi^{T_{f^*}}(\cdot|x)||\pi^{W^\epsilon}(\cdot|x))d\mathbb{P}_0(x) + \int_{\mathcal{X}} \log C_{\hat{\beta}}^x d\mathbb{P}_0(x) - \int \hat{\beta}(y)d\mathbb{P}_1(y) =$$

$$\int_{\mathcal{X}} \text{KL}(\pi^{T_{f^*}}(\cdot|x)||\pi^{W^\epsilon}(\cdot|x))d\mathbb{P}_0(x) + \int_{\mathcal{X}} \log C_{\hat{\beta}}^x d\mathbb{P}_0(x) - \int \hat{\beta}(y)d\pi^{T_{f^*}}(y|x)d\mathbb{P}_0(x) =$$

$$\int_{\mathcal{X}} \left\{ \text{KL}(\pi^{T_{f^*}}(\cdot|x)||\pi^{W^\epsilon}(\cdot|x)) + \log C_{\hat{\beta}}^x - \int_{\mathcal{Y}} \hat{\beta}(y)d\pi^{T_{f^*}}(y|x) \right\}d\mathbb{P}_0(x) =$$

$$\int_{\mathcal{X}} \left\{ \text{KL}(\pi^{T_{f^*}}(\cdot|x)||\pi^{T^{\hat{\beta}}}(\cdot|x)) - \log C_{\hat{\beta}}^x + \log C_{\hat{\beta}}^x \right\}d\mathbb{P}_0(x) =$$

$$\int_{\mathcal{X}} \text{KL}(\pi^{T_{f^*}}(\cdot|x)||\pi^{T^{\hat{\beta}}}(\cdot|x))d\mathbb{P}_0(x) = \text{KL}(\pi^{T_{f^*}}||\pi^{T^{\hat{\beta}}}) =$$

$$\text{KL}(\pi^{T_{f^*}}||\pi^{T^{\hat{\beta}}}) + \underbrace{\int_{\mathcal{X}\times\mathcal{Y}} \text{KL}(T_{f^*|x,y}||T_{|x,y}^{\hat{\beta}})d\pi^{T_{f^*}}(x,y)}_{=0, \text{ since } T_{f^*|x,y}=T_{|x,y}^{\hat{\beta}}=W_{|x,y}^\epsilon} = \text{KL}(T_{f^*}||T^{\hat{\beta}}). \quad (42)$$

Thus, we obtain $\epsilon_2 = \text{KL}(T_{f^*}||T^{\hat{\beta}})$.

Part 3. By summing (41) and (42) and using the Pinsker inequality, we obtain

$$\epsilon_1 + \epsilon_2 = \text{KL}(T_{\hat{f}}||T^{\hat{\beta}}) + \text{KL}(T_{f^*}||T^{\hat{\beta}}) \geq 2\rho_{\text{TV}}^2(T_{\hat{f}}, T^{\hat{\beta}}) + 2\rho_{\text{TV}}^2(T_{f^*}, T^{\hat{\beta}}) \geq$$
$$\left[ \rho_{\text{TV}}(T_{\hat{f}}, T^{\hat{\beta}}) + \rho_{\text{TV}}(T_{f^*}, T^{\hat{\beta}}) \right]^2 \geq \rho_{\text{TV}}^2(T_{\hat{f}}, T_{f^*}). \quad (43)$$

Here we use the triangle inequality in line (43). Therefore, $\rho_{\text{TV}}(T_{\hat{f}}, T_{f^*}) \leq \sqrt{\epsilon_1 + \epsilon_2}$.

Part 4. By summing (41) and (42) and using the Pinsker inequality, we obtain

$$\epsilon_1 + \epsilon_2 = \text{KL}(T_{\hat{f}}||T^{\hat{\beta}}) + \text{KL}(T_{f^*}||T^{\hat{\beta}}) =$$

$$\text{KL}(\pi^{T_{\hat{f}}}||\pi^{T^{\hat{\beta}}}) + \int_{\mathcal{X}\times\mathcal{Y}} \text{KL}(T_{\hat{f}|x,y}||T_{|x,y}^{\hat{\beta}})d\pi^{T_{\hat{f}}}(x,y) +$$

$$\text{KL}(\pi^{T_{f^*}}||\pi^{T^{\hat{\beta}}}) + \int_{\mathcal{X}\times\mathcal{Y}} \text{KL}(T_{f^*|x,y}||T_{|x,y}^{\hat{\beta}})d\pi^{T_{f^*}}(x,y) \geq$$

$$\text{KL}(\pi^{T_{\hat{f}}}||\pi^{T^{\hat{\beta}}}) + \text{KL}(\pi^{T_{f^*}}||\pi^{T^{\hat{\beta}}}) \geq 2\rho_{\text{TV}}^2(\pi^{T_{\hat{f}}}, \pi^{T^{\hat{\beta}}}) + 2\rho_{\text{TV}}^2(\pi^{T_{f^*}}, \pi^{T^{\hat{\beta}}}) \geq$$

$$\left[ \rho_{\text{TV}}(\pi^{T_{\hat{f}}}, \pi^{T^{\hat{\beta}}}) + \rho_{\text{TV}}(\pi^{T_{f^*}}, \pi^{T^{\hat{\beta}}}) \right]^2 \geq \rho_{\text{TV}}^2(\pi^{T_{\hat{f}}}, \pi^{T_{f^*}}).$$

Thus, $\rho_{\text{TV}}(\pi^{T_{f^*}}, \pi^{T^{\hat{\beta}}}) \leq \sqrt{\epsilon_1 + \epsilon_2}$. $\qquad \square$

## C Euler-Maruyama

In our Algorithm 1, at both the training and the inference stages, we use the Euler-Maruyama Algorithm 2 to solve SDE.

## D Drift Norm Constant Multiplication Invariance

Our Algorithm 1 aims to solve the following optimization problem:

$$\sup_\beta \inf_{T_f \in \mathcal{D}(\mathbb{P}_0)} \underbrace{\left\{ \mathbb{E}_{T_f}\left[\int_0^1 C||f(X_t, t)||^2 dt\right] + \int_{\mathcal{Y}} \beta(y)d\mathbb{P}_1(y) - \int_{\mathcal{Y}} \beta(y)d\mathbb{P}_1^{T_f}(y) \right\}}_{\overset{\text{def}}{=}\mathcal{L}^C(\beta, T_f)},$$

with $C = 1$. At the same time, we use $C = \frac{1}{2\epsilon}$ in our theoretical derivations (12). We emphasize that the actual value of $C > 0$ **does not affect** the optimal solution $T_{f^*}$ to this problem. Specifically, if

---

**Algorithm 2:** Euler-Maruyama algorithm

---

**Input** : batch of initial states $X_0$ at time moment $t = 0$;
           SDE drift network $f_\theta : \mathbb{R}^D \times [0, 1] \to \mathbb{R}^D$;
           number of steps for the SDE solver $N \geq 1$;
           noise variance $\epsilon \geq 0$.

**Output** : batches $\{X_n\}_{n=0}^N$ of intermediate states at $t = \frac{n}{N}$ simulating the proccess
           $dX_t = f(X_t, t)dt + \sqrt{\epsilon}dW_t$;
           batches $\{f_n\}_{n=0}^N$ of drift values $f(X_n, t_n)$ at $t = \frac{n-1}{N}$ simulating the process;

$\Delta t \leftarrow \frac{1}{N}$ ;
**for** $t = 1, 2, \dots, N$ **do**
     **for** $i = 1, 2, \dots, |X_0|$ **do**
         Sample noise $W$ from $\mathcal{N}(0, I)$ ;
         $f_{t-1,i} \leftarrow f(X_{t-1}, t-1)$ ;
         $X_{t,i} \leftarrow X_{t-1,i} + f_{t-1,i}\Delta t + \sqrt{\epsilon \Delta t}W$ ;

---

$(\beta^*, T_{f^*})$ is the optimal point for the problem with $C = 1$, then $(\widetilde{C}\beta^*, T_{f^*})$ is the optimal point for $C = \widetilde{C}$. Indeed, for a pair $(\beta, T_f)$ it holds that

$$\mathcal{L}^1(\beta, T_f) = \mathbb{E}_{T_f}\Big[\int_0^1 ||f(X_t, t)||^2 dt\Big] + \int_{\mathcal{Y}} \beta(y)d\mathbb{P}_1(y) - \int_{\mathcal{Y}} \beta(y)d\mathbb{P}_1^{T_f}(y) =$$

$$\frac{1}{\widetilde{C}}\Big\{\mathbb{E}_{T_f}\Big[\int_0^1 \widetilde{C}||f(X_t, t)||^2 dt\Big] + \int_{\mathcal{Y}} \widetilde{C}\beta(y)d\mathbb{P}_1(y) - \int_{\mathcal{Y}} \widetilde{C}\beta(y)d\mathbb{P}_1^{T_f}(y)\Big\} =$$

$$\frac{1}{\widetilde{C}}\Big\{\mathbb{E}_{T_f}\Big[\int_0^1 \widetilde{C}||f(X_t, t)||^2 dt\Big] + \int_{\mathcal{Y}} \widetilde{\beta}(y)d\mathbb{P}_1(y) - \int_{\mathcal{Y}} \widetilde{\beta}(y)d\mathbb{P}_1^{T_f}(y)\Big\} = \frac{1}{\widetilde{C}}\mathcal{L}^{\widetilde{C}}(\widetilde{\beta}, T_f), \quad (44)$$

where we use $\widetilde{\beta} \stackrel{\text{def}}{=} \widetilde{C}\beta$. Hence problems $\sup_\beta \inf_{T_f} \mathcal{L}^1(\beta, T_f)$ and $\sup_{\widetilde{\beta}} \inf_{T_f} \mathcal{L}^{\widetilde{C}}(\widetilde{\beta}, T_f)$ can be viewed as **equivalent** in the sense that one can be derived one from the other via the change of variables and multiplication by $\widetilde{C} > 0$. For completeness, we also note that the change of variables $\beta \leftrightarrow \widetilde{\beta}$ actually preserves the functional class of $\beta$, i.e., $\beta \in \mathcal{C}_{b,2}(\mathcal{Y}) \iff \widetilde{\beta} \in \mathcal{C}_{b,2}(\mathcal{Y})$.

For convenience, we get rid of dependence on $\epsilon$ in the objective (12) and consider $\mathcal{L}^1$ for optimization, i.e., use $C = 1$ in Algorithm 1. Still the dependence on $\epsilon$ remains in $\sup_\beta \inf_{T_f} \mathcal{L}^1(\beta, T_f)$ as $T_f \in \mathcal{D}(\mathbb{P}_0)$ is a diffusion process with volatility $\epsilon$. Interestingly, this point of view (optimizing $\mathcal{L}^1$ instead of $\mathcal{L}^{\frac{1}{2\epsilon}}$) **technically** allows to consider even $\epsilon = 0$. In this case, the optimization is performed over *deterministic* trajectories $T_f$ determined by the velocity field $f(X_t, t)$. The problem $\sup_\beta \inf_{T_f} \mathcal{L}^1(\beta, T_f)$ may be viewed as a saddle point reformulation of the **unregularized** OT with the quadratic cost in the *dynamic* form, also known as the Benamou-Brenier formula [43, §6.1]. This particular case is out of scope of our paper (it is not EOT/SB) and we do not study the properties of $\mathcal{L}^1$ in this case. However, for completeness, we provide experimental results for $\epsilon = 0$.

## E ENOT for Toy Experiments and High-dimensional Gaussians

In 2D toy experiments, we consider 2 tasks: *Gaussian → 8 gaussians* and *Gaussian → Swiss roll*. Results for the last one (Figure 5) are qualitatively similar to results of the first one (Figure 2), which we discussed earlier (§5.1). For both tasks, we parametrize the SDE drift function in Algorithm 1 by a feedforward neural network $f_\theta$ with 3 inputs, 3 linear layers (100 hidden neurons and ReLU activations) and 2 outputs. As inputs, we use 2 coordinates and time value $t$ (as is). Analogically, we parametrize the potential by a feedforward neural network $\beta_\phi$ with 2 inputs, 3 linear layers (100 hidden neural and ReLU activations) and 2 outputs. In all the cases, we use $N = 10$ discretization steps for solving SDE by Euler-Maruyama Algorithm 2, Adam with lr $= 10^{-4}$, batch size 512. We train the model for 20000 total iterations of $\beta_\phi$, and on each of them, we do $K_f = 10$ updates for the SDE drift function $f_\theta$.

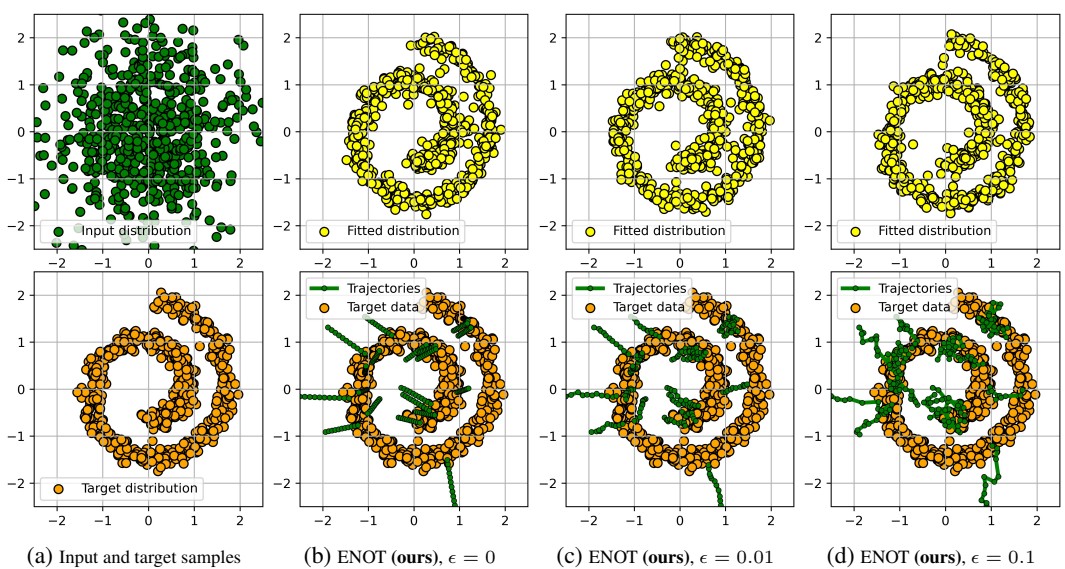

Figure 5: Gaussian → Swiss roll, learned stochastic process with ENOT **(ours)**.

(a) Input and target samples  (b) ENOT **(ours)**, $\epsilon = 0$  (c) ENOT **(ours)**, $\epsilon = 0.01$  (d) ENOT **(ours)**, $\epsilon = 0.1$

In the experiments with high-dimensional Gaussians, we use exactly the same setup as for toy 2D experiments but chose $N = 200$ discretization steps for SDE, all hidden sizes in neural networks are 512, and we train our model for 10000 iterations. To illustrate the stability of the algorithm, we provide the plot of $\text{BW}_2^2$-UVP (%) between the ground truth EOT plan $\pi^*$ and the learned plan $\pi$ of ENOT during training for $DIM = 128$ in Figure 6.

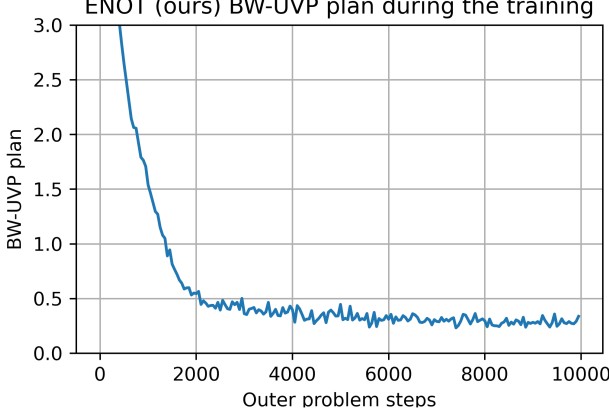

Figure 6: $\text{BW}_2^2$-UVP ↓ (%) between the the EOT plan $\pi^*$ and the learned plan $\pi$ of ENOT and MLE-SB during the training (DIM = 128).

## F ENOT for Colored MNIST and Unpaired Super-resolution of Celeba Faces

For the image tasks (§5.3, §5.4), we find out that using the following reparametrization of Euler-Maruyama Algorithm 2 considerably improves the quality of our Algorithm 1. In the Euler-Maruyama Algorithm 2, instead of using a neural network to parametrize drift function $f(X_t, t)$ and calculating the next state as $X_{t+1} = X_t + f(X_t, t)\Delta t + \sqrt{\epsilon \Delta t}$, we parametrize $g(X_t, t) = X_t + f(X_t, t)\Delta t$ by a neural network $g_\theta$, and calculate the next state as $X_{t+1} = g_\theta(X_t, t) + \sqrt{\epsilon \Delta t}$. In turn, the drift function is given by $f(X_t, t) = \frac{1}{\Delta t} g(X_t, t) - X_t$. Also, we do not add noise at the last step of the Euler-Maruyama simulation because we find out that it provides better empirical performance.

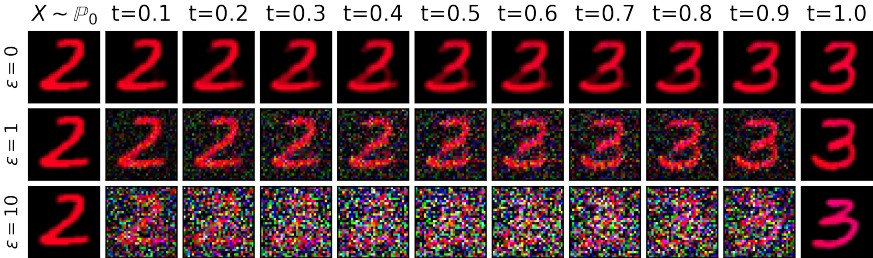

Figure 7: Trajectories from our learned ENOT **(ours)** for colored MNIST for different $\epsilon$.

We use WGAN-QC discriminator's ResNet architecture [5] for the potential $\beta$. We use UNet [6] as $g_\theta(X_t, t)$ of SDE in our model. To condition it on $t$, we first obtain the embedding of $t$ by using the positional embedding [7]. Then we add conditional instance normalization (CondIN) layers after each UNet's upscaling block [8]. We use Adam with lr $= 10^{-4}$, batch size 64 and 10:1 update ratio for $f_\theta/\beta_\phi$. For $\epsilon = 0$ and $\epsilon = 1$ our model converges in $\approx 20000$ iterations, while for $\epsilon = 10$ it takes $\approx 70000$ iteration to convergence. The last setup takes more iterations to converge because adding noise with higher variance during solving SDE by Euler-Maruyama Algorithm 2 increases the variance of stochastic gradients.

In the unpaired super-resolution of Celeba faces, we use the same experimental setup as for the colored MNIST experiment. It takes $\approx 40000$ iterations for $\epsilon = 0$ and $\approx 70000$ iterations for $\epsilon = 1$ and $\epsilon = 10$ to converge. In Figures 7, 1 we present trajectories provided by our algorithm for Colored MNIST and Celeba experiments.

**Computational complexity.** In the most challenging task (§5.4), ENOT converges in one week on $2\times$ A100 GPUs.

# G   Details of the baseline methods

In this section, we discuss details of the baseline methods with which we compare our method.

### G.1   Gaussian case (§5.2).

**SCONES** [14]. We use the code from the authors' repository



`https://github.com/mdnls/scones-synthetic`



for their evaluation in the Gaussian case. We employ their configuration `blob/main/config.py`.

**LSOT** [45]. We use the part of the code of SCONES corresponding to learning dual OT potentials `blob/main/cpat.py` and the barycentric projection `blob/main/bproj.py` in the Gaussian case with configuration `blob/main/config.py`.

**FB-SDE-J** [10]. We utilize the official code from



`https://github.com/ghliu/SB-FBSDE`



with their configuration `blob/main/configs/default_checkerboard_config.py` for the checkerboard-to-noise toy experiment, changing the number of steps of dynamics from 100 to 200 steps. Since their hyper-parameters are developed for their 2-dimensional experiments, we increase the number of iterations for dimensions 16, 64 and 128 to 15 000.

**FB-SDE-A** [10]. We also take the code from the same repository as above. We base our configuration on the authors' one (`blob/main/configs/default_moon_to_spiral_config.py`) for the moon-to-spiral experiment. As earlier, we increase the number of steps of dynamics up to 200. Also, we change the number of training epochs for dimensions 16, 64 and 128 to 2,4 and 8 correspondingly.

---

[5] github.com/harryliew/WGAN-QC
[6] github.com/milesial/Pytorch-UNet
[7] github.com/rosinality/denoising-diffusion-pytorch
[8] github.com/kgkgzrtk/cUNet-Pytorch

**DiffSB [15]**. We utilize the official code from

```
https://github.com/JTT94/diffusion_schrodinger_bridge
```

with their configuration `blob/main/conf/dataset/2d.yaml` for toy problems. We increase the amount of steps of dynamics to 200 and the number of steps of IPF procedure for dimensions 16, 64 and 128 to 30, 40 and 60, respectively.

**MLE-SB [48]**. We use the official code from

```
https://github.com/franciscovargas/GP_Sinkhorn
```

with hyper-parameters from `blob/main/notebooks/2D Toy Data/2d_examples.ipynb`. We set the number of steps to 200. As earlier, we increase the number of steps of IPF procedure for dimensions 16, 64 and 128 to 1000, 3500 and 5000, respectively.

## G.2 Colored MNIST (§5.3)

**SCONES [14]**. In order to prepare a score-based model, we use the code from

```
https://github.com/ermongroup/ncsnv2
```

with their configuration `blob/master/configs/cifar10.yml`. Next, we utilize the code of SCONES from the official repository for their unpaired Celeba super-resolution experiment (`blob/main/scones/configs/superres_KL_0.005.yml`). We adapt it for $32\times32$ ColorMNIST images instead of $64\times64$ celebrity faces.

**DiffSB [15]**. We use the official code with their configuration `blob/main/conf/mnist.yaml` adopting it for three-channel ColorMNIST images instead of one-channel MNIST digits.

## G.3 CelebA (§5.4)

**SCONES [14]**. For the SCONES, we use their exact code and configuration from `blob/main/scones/configs/superres_KL_0.005.yml`. As for the score-based model for celebrity faces, we pick the pre-trained model from

```
https://github.com/ermongroup/ncsnv2
```

It is the one used by the authors of SCONES in their paper.

**Augmented Cycle GAN [2]**. We use the official code from

```
https://github.com/NathanDeMaria/AugmentedCycleGAN
```

with their default hyper-parameters.

**ICNN [38]**. We utilize the reworked implementation by

```
https://github.com/iamalexkorotin/Wasserstein2Benchmark.
```

which is a non-minimax version [26] of ICNN-based approach [38]. That is, we use `blob/main/notebooks/W2_test_images_benchmark.ipynb` and only change the dataloaders.

## H Mean-Field Games

This appendix discusses the relation between the Mean-Field Game problem and Schrödinger Brdiges.

### H.1 Intro to the Mean-Field game.

Consider a game with infinitely many small players. At time moment $t = 0$, they are distributed according to $X_0 \sim \rho_0$. Every player controls its behavior through drift $\alpha$ of the SDE:

$$dX_t = \alpha(X_t, t, \rho_t)dt + \sqrt{2\nu}dW_t$$

Here $\rho_t$ is the distribution of all the players at the time moment $t$. When we consider a specific player, we consider $\rho_t$ as a parameter. Each player aims to minimize the quantity:

$$\mathbb{E}[\int_0^T (L(X_t, \alpha_t, \rho_t) + f(X_t, \rho_t))dt + g(X_T, \rho_T)].$$

Here $L(x, \alpha, \rho)$ is similar to the Lagrange function in physics and describes the cost of moving in some direction given the current position and the other players' distribution. The additional function $f(X_t, \rho_t)$ is interpreted as the cost of the player's interaction at coordinate $x$ with all the others. Now we can introduce the value function $\phi(x, t)$, which for position $x$ and start time $t$ returns the cost in case of the optimal control:

$$\phi(x, t) \stackrel{\text{def}}{=} \inf_\alpha \mathbb{E}[\int_t^T (L(X_t, \alpha_t, \rho_t) + f(X_t, \rho_t))dt + g(X_T, \rho_T)].$$

Before considering the Mean-Field game, we need to define an additional function $H(x, p, \rho)$. It is similar to the Hamilton function and is defined as the Legendre transform of Lagrange function $L$:

$$H(x, p, \rho) \stackrel{\text{def}}{=} \sup_\alpha [-\alpha p - L(x, \alpha, \rho)].$$

*Mean-Field game implies finding the Nash equilibrium for all players of such the game.* It is known [1] that the Nash equilibrium is the solution of the system of Hamilton-Jacobi-Bellman (HJB) and Fokker-Planck (FP) PDE equations. For two functions $H(x, p, \rho)$ and $f(x, \rho)$, Mean-Field game formulates as a system of two PDE with two constraints:

$$-\partial_t \phi - \nu \Delta \phi + H(x, \nabla \phi, \rho) = f(x, \rho) \text{ (HJB)}$$

$$-\partial_t \rho - \nu \Delta \rho - \mathbf{div}(\rho \nabla_p H(x, \nabla \phi)) = 0 \text{ (FP)}$$

$$\text{s.t. } \rho(x, 0) = \rho_0 \text{ , } \phi(x, T) = g(x, \rho(\cdot, T))$$

The solution of this system is two functions $\rho(x, t)$ and $\phi(x, t)$, which describe all players' dynamics. Also, in Nash equilibrium, the specific player's behavior is described by the following SDE:

$$dX_t = -\nabla_p H(X_t, \nabla \phi(X_t, t), \rho)dt + \sqrt{2\nu}dW_t.$$

## H.2 Relation to our work.

In recent work [34], the authors show that the Schrodinger Bridger problem could be formulated as a Mean-Field game with hard constraints on distribution $\rho(\cdot, T) = \rho_{target}(\cdot, T)$ via choosing proper function $g(x, \rho(\cdot, T))$ such as:

$$g(x, \rho(\cdot, T)) = \begin{cases} \infty, & \text{if } \rho(\cdot, T) \neq \rho_{target}(\cdot, T) \\ 0, & \rho(\cdot, T) = \rho_{target}(\cdot, T) \end{cases}$$

Also, the authors proposed an extension of DiffSB [34] algorithm for the Mean-Field game problem.

In [33], the authors in their experiments **consider only soft constraints** on the target density. More precisely, they consider only simple constraints such as $g(x, \rho) = ||x - x_{target}||_2$, where $x_{target}$ is a given shared target point for every player, and every player is penalized for being far from this. Such soft constraint force players to have delta distribution at point $x_{target}$.

To solve the Mean-Field problem, the authors parameterize value function $\phi(x, t)$ by a neural network and use different neural network $N_\theta$ to sample from $\rho_t$. The authors penalize the violation of Mean-Field game PDEs for optimizing these networks. After the convergence, one can sample from the distribution $\rho_t$ by using neural network $N_\theta$. *Approach [33] has the advantage that authors do not need to use SDE solvers, which require more steps with growing parameter $\nu$ of diffusion operator.* However, computation of Laplacian and divergence for high-dimensional spaces (e.g., space 12228-dimensional space of 3x64x64 images) at each iteration of the training step may be computationally hard, restricting the applicability of their method to large-scale setups.

**In our approach**, we initially work with the SDE:

$$dX_t = \alpha(X_t, t, \rho_t)dt + \sqrt{2\nu}dW_t,$$

which describes the player's behavior and use a neural network to parametrize the drift $\alpha$. We consider only **hard constraints** on the target distribution, $f(X_t, \rho_t) = 0$ and $L(X_t, \alpha_t, \rho_t) = \frac{1}{2}||\alpha_t||^2$ since this variant of Mean-Field game is also the particular case of Schrodinger Bridge problem and is equivalent to the entropic optimal transport. *Since we do not need to compute Laplacian or divergence, our approach scales better with the dimension.* However, for high values of diffusion parameter $\nu$ (which is equal to the $\frac{1}{2}\epsilon$ in our notation, where $\epsilon$ is the entropic regularization strength), our approach needs more steps for accurate solving of the SDE to provide samples, as we mentioned in limitations.

# I   Extending ENOT to other costs

In the main text, we focus only on EOT with the quadratic cost $c(x, y) = \frac{1}{2}||x - y||^2$ which coincides with SB with the Wiener prior $W^\epsilon$. However, one could use a different prior $Q_v$ instead of $W^\epsilon$ in (5):

$$Q_v : dX_t = v(X_t, t)dt + \sqrt{\epsilon}dW_t,$$

and solve the problem

$$\inf_{T_f \in \mathcal{D}(\mathbb{P}_0, \mathbb{P}_1)} \text{KL}(T_f || Q_v) = \inf_{T_f \in \mathcal{D}(\mathbb{P}_0, \mathbb{P}_1)} \frac{1}{2\epsilon}\mathbb{E}_{T_f}\left[\int_0^1 ||f(X_t, t) - v(X_t, t)||^2 dt\right].$$

Here we just use the known expression (6) for $\text{KL}(T_f || Q_v)$ between two diffusion processes through their drift functions. Using the same derivation as in the main text §2.2, it can be shown that this new problem is equivalent to solving the EOT with cost $c(x, y) = -\log \pi^{Q_v}(y|x)$, where $\pi^{Q_v}(y|x)$ is a conditional distribution of the stochastic process $Q_v$ at time $t = 1$ given the starting point $x$ at time $t = 0$. For example, for $W^\epsilon$ (which we consider in the main text) we have

$$c(x, y) = -\log \pi^{W^\epsilon}(y|x) = \frac{1}{2\epsilon}(y - x)^T(y - x) + \text{Const},$$

i.e., we get the quadratic cost. Thus, using different priors for the Schrodinger bridge problem makes it possible to solve Entropic OT for other costs. We conjecture that most of our proofs and derivations can be extended to arbitrary prior process $Q_v$ just by slightly changing the minimax functional (12):

$$\sup_\beta \inf_{T_f} \left( \frac{1}{2\epsilon}\mathbb{E}_{T_f}\left[\int_0^1 ||f(X_t, t) - v(X_t, t)||^2 dt\right] + \int_{\mathcal{Y}} \beta_\phi(y)d\mathbb{P}_1(y) - \int_{\mathcal{Y}} \beta_\phi(y)d\pi_1^{T_f}(y) \right).$$

We conduct a toy experiment to support this claim and consider $Q_v$ with $\epsilon = 0.01$ and $v(x, t) = \nabla \log p(x)$, where $\log p(x)$ is a 2D distribution with a wave shape, see Figure 8. Intuitively, it means that trajectories should be concentrated in the regions with a high density of $p$. In Figure e 8, there the grey-scale color map represents the density of $p$, start points ($\mathbb{P}_0$) are green, target points ($\mathbb{P}_1$) are red, obtained trajectories are pink and mapped points are blue.

# J   ENOT for the unregularized OT ($\epsilon = 0$)

Our proposed algorithm is designed to solve entropic OT and the equivalent SB problem. This implies that $\epsilon > 0$. Nevertheless, our algorithm *technically* allows using even $\epsilon = 0$, in which case it presumably computes the unegularized OT map for the quadratic cost. Here we present some empirical evidence supporting this claim as well some theoretical insights.

EMPIRICAL EVIDENCE. We consider the experimental setup with images from the continuous Wasserstein-2 benchmark [28, §4.4]. The images benchmark provides 3 pairs of distributions (Early, Mid, Late) for which the ground truth unregularized OT map for the quadratic cost is known by the construction. Hence, we may compare the map learned with our method ($\epsilon = 0$) with the true one.

We train our method with $\epsilon = 0$ on each of 3 benchmark pairs and present the quantitative results in Table 6. We use the same $\mathcal{L}^2$-UVP metric [28, §4.2] as the authors of the benchmark. As the baselines, we include the results of MM:R method from [28] and the method from [3]. Both methods are minimax and have some similarities with our approach. As we can see, ENOT with $\epsilon = 0$

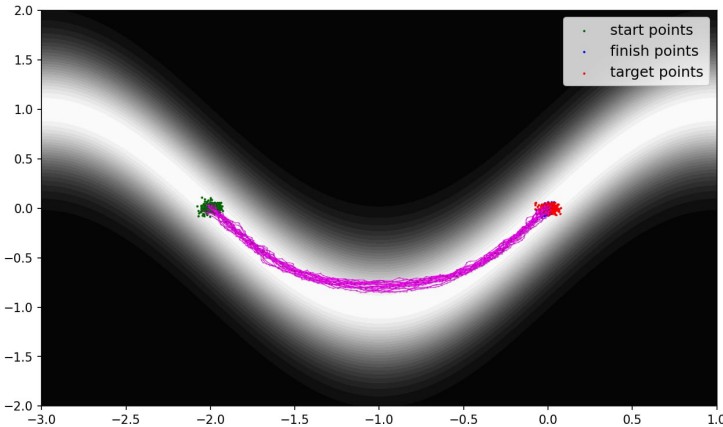

Figure 8: Toy example with ENOT (**ours**) for the complex prior $Q_v : dX_t = v(X_t, t)dt + \sqrt{\epsilon}dW_t$.

| Benchmark | Early | Mid | Late |
|-----------|-------|-----|------|
| [28]* | 1.4 | 0.4 | 0.22 |
| [3]* | 0.61 | 0.20 | 0.09 |
| ENOT (**ours**) | 0.77 | 0.21 | 0.09 |

Table 6: Comparison on W2 benchmark. *Results are taken from [3, Table 2].

works better than the MM:R solver but slightly underperforms compared to [3]. This evaluation demonstrates that our method recovers the unregularized OT map for the quadratic cost with the comparable quality to the existing saddle point OT methods.

THEORETICAL INSIGHTS. We see that empirically our method with $\epsilon = 0$ recovers the unregularized OT map. At the same time, this is not supported by our theoretical results as they work exclusively for $\epsilon > 0$ and rely on the properties of the KL divergence.

Overall, it seems like for $\epsilon = 0$ our method yields a saddle point reformulation of the Benamou-Brenier (BB) [8] problem which is also known as the dynamic version of the unregularized OT ($\epsilon = 0$) with the quadratic cost. This problem can be formulated as follows:

$$\inf_{T_f} \left\{ \frac{1}{2}\mathbb{E}_{T_f}[\int_0^1 ||f(X_t,t)||^2 dt] \right\} \quad \text{s.t.} \quad T_f : dX_t = f(X_t,t)dt, \quad X_0 \sim \mathbb{P}_0, X_1 \sim \mathbb{P}_1, \quad (45)$$

i.e., the goal is to find the process $T_f$ of the minimal energy which moves the probability mass of $\mathbb{P}_0$ to $\mathbb{P}_1$. BB (45) is very similar to DSB (11) but there is no multiplier $\frac{1}{\epsilon}$, and the stochastic process $T_f$ is restricted to be deterministic ($\epsilon = 0$). It is governed by a vector field $f$. Just like the DSB (11) is equivalent to EOT (2), it is known that BB (45) is **equivalent** to unregularized OT with the quadratic cost ($\epsilon = 0$). Namely, the distribution $\pi^{T_{f*}}$ is the unregularized OT plan between $\mathbb{P}_0$ and $\mathbb{P}_1$.

In turn, our Algorithm 1 for $\epsilon = 0$ optimizes the following saddle point objective:

$$\sup_{\beta} \inf_{T_f} \mathcal{L}(\beta, T_f) \stackrel{\text{def}}{=} \sup_{\beta} \inf_{T_f} \left\{ \frac{1}{2}\mathbb{E}_{T_f}[\int_0^1 ||f(X_t,t)||^2 dt] + \int_{\mathcal{Y}} \beta(y)d\mathbb{P}_1(y) - \int_{\mathcal{Y}} \beta(y)d\mathbb{P}_1^{T_f}(y) \right\}, \quad (46)$$

where $T_f : dX_t = f(X_t,t)dt$ with $X_0 \sim \mathbb{P}_0$ (the constraint $X_1 \sim \mathbb{P}_1$ here is lifted) and $\beta \in \mathcal{C}_{2,b}(\mathcal{Y})$. Just like in the Entropic case, functional $\mathcal{L}$ can be viewed as the Lagrangian for BB (45) with $\beta$ playing the role of the Lagrange multiplier for the constraint $d\pi_1^{T_f}(y) = d\mathbb{P}_1(y)$. Naturally, it is expected that the value (45) coincides with (46), and we provide a *sketch of the proof* of this fact.

Overall, the proof logic is analogous to the Entropic case but the actual proof is much more technical as we can not use the $KL$-divergence machinery which helps to avoid non-uniqueness, etc.

**Step 1 (Auxiliary functional, analog of Lemma B.3).** We introduce an auxiliary functional

$$\widetilde{\mathcal{L}}(\beta, H) \stackrel{\text{def}}{=} \int_{\mathcal{X}} \frac{1}{2}||x - H(x)||^2 d\mathbb{P}_0(x) - \int_{\mathcal{X}} \beta(H(x))d\mathbb{P}_0(x) + \int_{\mathcal{Y}} \beta(y)d\mathbb{P}_1(y),$$

where $\beta$ is a potential and $H : \mathbb{R}^D \rightarrow \mathbb{R}^D$ is a measurable map. This functional is nothing but the well-known max-min reformulation of static OT problem (in Monge's form) with the quadratic cost [3, Eq. 4], [28, Eq.9]. Hence,

$$\sup_{\beta} \inf_{H} \widetilde{\mathcal{L}}(\beta, H) = \underbrace{\inf_{H\sharp\mathbb{P}_0=\mathbb{P}_1} \int_{\mathcal{X}} \frac{1}{2}||x - H(x)||^2 d\mathbb{P}_0(x)}_{\overset{\text{def}}{=}\mathcal{L}^*}.$$

**Step 2 (Solution of the inner problem is always an OT map).** An existence of some minimizer $H = H^\beta$ in $\inf_H \widetilde{L}(\beta, H)$ can be deduced from the measurable argmin selection theorem, e.g., [21, Theorem 18.19]. For this $H^\beta$ we consider $\mathbb{P}' \overset{def}{=} H^\beta\sharp\mathbb{P}_0$. Recall that

$$H^\beta \in \underset{H}{\arg\inf} \, \widetilde{L}(\beta, H) = \underset{H}{\arg\inf} \int_{\mathcal{X}} \big\{ \frac{||x - H(x)||^2}{2} - \beta(H(x)) \big\} d\mathbb{P}_0(x).$$

Here we may add the fictive constraint $H\sharp\mathbb{P}_0 = \mathbb{P}'$ which is anyway satisfied by $H^\beta$ and get

$$H^\beta \in \underset{H\sharp\mathbb{P}_0=\mathbb{P}'}{\arg\inf} \int_{\mathcal{X}} \big\{ \frac{||x - H(x)||^2}{2} - \beta(H(x)) \big\} d\mathbb{P}_0(x) = \underset{H\sharp\mathbb{P}_0=\mathbb{P}'}{\arg\inf} \int_{\mathcal{X}} \frac{||x - H(x)||^2}{2} d\mathbb{P}_0(x).$$

The last equality holds since $\int \beta(H(x))d\mathbb{P}_0(x) = \int \beta(y)d\mathbb{P}'(y)$ does not depend on the choice of $H$ due to the constraint $H\sharp\mathbb{P}_0 = \mathbb{P}'$. The latter is the OT problem between $\mathbb{P}$ and $\mathbb{P}'$ and we see that $H^\beta$ is its solution.

**Step 3 (Equivalence for inner objective values).** Since $H^\beta$ is the OT map between $\mathbb{P}_0, \mathbb{P}'$ (it is unique as $\mathbb{P}_0$ is absolutely continuous [43]), it can be represented as an ODE solution $T_{f^\beta}$ to the Benamour Brenier problem between $\mathbb{P}_0, \mathbb{P}'$, i.e., $T_{f^\beta} : dX_t = f^\beta(X_t, t)dt$ and $H^\beta(X_0) = X_0 + \int_0^1 f^\beta(X_t, t)dt$. Furthermore, in this case, $||X_0 - H^\beta(X_0)||^2 = \int_0^1 ||f^\beta(X_t, t)||^2dt$. Hence, it can be derived that

$$\inf_{H} \widetilde{\mathcal{L}}(\beta, H) = \inf_{T_f} \mathcal{L}(\beta, T_f).$$

**Step 4 (Equivalence of the saddle point objective).** Take sup over $\beta \in \mathcal{C}_{b,2}(\mathcal{Y})$ and get the final equivalence:

$$\sup_{\beta} \inf_{H} \widetilde{\mathcal{L}}(\beta, H) = \sup_{\beta} \inf_{H} \mathcal{L}(\beta, T_f) = \mathcal{L}^*.$$

**Step 5 (Dynamic OT solutions are contained in optimal saddle points).** Pick any optimal $\beta^* \in \arg\sup_{\beta} \inf_H \mathcal{L}(\beta, T_{f^*})$ and let $T_{f^*}$ be any solution to the Benamou-Brenier problem. Checking that $T^* \in \inf_H \mathcal{L}(\beta^*, T_f)$ can be done analogously to [31, Lemma 4], [42, Lemma 4.1]. $\square$

The derivation above shows the equivalence of objective values of dynamic unregularized OT (45) and our saddle point reformulation of BB (45). Additionally, it shows that solutions $T_{f^*}$ can be recovered from *some* optimal saddle points $(\beta^*, T_{f^*})$ of our problem. At the same time, *unlike the EOT case* ($\epsilon > 0$), it is not guaranteed that for all the optimal saddle points $(\beta^*, T_{f^*})$ it holds that $T_{f^*}$ is the solution to the BB problem. This aspect seems to be closely related to the *fake solutions* issue in the saddle point methods of OT [30] and may require further studies.

