# OpenReview forum: "Entropic Neural Optimal Transport via Diffusion Processes"
_NeurIPS.cc/2023/Conference — NeurIPS 2023 oral_

### Official Review · Reviewer_KyaP · 2023-06-30

**Soundness:** 3 good
**Presentation:** 2 fair
**Contribution:** 2 fair
**Rating:** 7
**Confidence:** 4

**Summary:**

The paper proposes to solve dynamic entropic optimal tansport (EOT), also known as Schrödinger bridge problem, with nerual solver. Specifically, the authors propose a saddle-point, maximin, formulation of EOT, yielding a GAN-resemble algorithm that can be trained in an end-to-end fashion. Experiments are conducted on 2D toy datasets and images translation.

**Strengths:**

- The saddle point reformulation of EOT is interesting. The proposed algorithm can be viewed as a stochastic control problem, where the terminal cost function $\beta$ is learned so that the resulting policy approaches terminal distribution.
- Writing is generally clear and easy to follow. Sufficient related works and preliminaries are included.
- Experiment are extensive and include many related baselines.

**Weaknesses:**

- The proposed algorithm aims to minimize (12), which is not well-defined when $\epsilon$=0, as the KL term will blow up. Even though $\epsilon$=0 is algorithmically applicable, it makes the current algorithm disconnected from the mathematical framework. I'll be more convinced if the authors can provide additional justification (maybe connection to OT).

- The proposed method is closely related to recent maximim OT [1], which consists of the same two networks (potential + policy) and the same training losses. From my understanding, the two methods coincide when $\epsilon$=0 and $N=1$. Are the authors aware of [1]? Given that the proposed ENOT seems to work best when $\epsilon$=0, can the author compare with [1]?

- Given that DiffSB [13] was compared throughout most Sec 5, I suggest the authors to compare in Sec 5.4 to [2], which applies DiffSB to unpaired super-resolution image datasets.

- While Sec 2 has introduced sufficient background and comparison between EOT and SB, which I do appreciate, I think their connection to Sec 4, which is the main content, is rather weak. Given that the proposed algorithm closely relates to dual formulation (e.g. (26) in Appendix), I suggest including those parts in Sec 2.

[1] On amortizing convex conjugates for optimal transport
[2] Conditional Simulation Using Diffusion Schrödinger Bridges


**Questions:**

- Experiments seem to support that $\epsilon$=0, where the dynamics reduce to flow, works much better than SDE. This seems to contradict recent insights from diffusion models [1], where SDE usually performs better than ODE on the same tasks. Can the authors comment on that?
- why is there a "1/|f|" term in Alg 1, when computing the KL? And what's $f_{n,m}$? I understand tht $f_n$ is the drift output at time step $n$.

[1] Elucidating the Design Space of Diffusion-Based Generative Models

**Limitations:**

Limitations are included in Sec6, where the authors mentioned the potential computational burden caused by simulation and back-prop through the SDE dynamics during training.

---

> ### Author Rebuttal · Authors · 2023-08-09
>
> Dear Reviewer KyaP, thank you for your comments. Here are the answers to your questions.
>
> **(1) Connection to OT for unregulated case $\epsilon=0$.**
>
> **We emphasize that our work focuses on developing a new algorithm for solving entropic OT and the equivalent SB problem. This implies that $\epsilon>0$.** We present the empirical results of our algorithm for $\epsilon=0$ only for completeness since it *technically* allows the use of $\epsilon=0$ (see lines 226-227). Given your interest in studying this case in detail, we provide **the proof sketch** to show that for $\epsilon=0$ our saddle-point objective
> $$
> \sup_{\beta} \inf_{T_f} \mathcal{L}(\beta,T_f) = \mathbb{E}_{T\_f}( \int\_0^1 ||f(X_t, t)||^2 dt) - \int\_{\mathcal{Y}} \beta(y) d\pi_1^{T_f}(y)+ \int\_{\mathcal{Y}} \beta(y) d\mathbb{P}_1(y) ,
> $$
>
> $$
> s.t. \quad X_0 \sim \mathbb{P}_0, \quad T_f : dX_t = f(X_t, t)dt.
> $$
> is a max-min reformulation of the **Benamou-Brenier** problem (the dynamic OT with $\ell^{2}$ cost):
> $$
> \mathcal{L}^*=\inf_f \mathbb{E}\_{T_f}[\int_0^1 ||f(X\_t, t)||^2 dt], \quad s.t. \quad X\_0 \sim \mathbb{P}\_0, \quad  T\_{f} : dX_t = f(X_t, t)dt, \quad X_1 \sim \mathbb{P}_1.
> $$
> which searches for an ODE drift $f$ which moves the mass of $\mathbb{P}_0$ to $\mathbb{P}_1$.
>
> **SKETCH OF THE PROOF**:
>
> *Step 1 (Auxiliary functional, analog of Lemma B.3).*
>
> We introduce an auxiliary functional:
> $$\widetilde{\mathcal{L}}(\beta,H)=\int\_{\mathcal{X}} \|x-H(x)\|^{2}d\mathbb{P}\_{0}(x)-\int\_{\mathcal{X}} \beta(H(x))d\mathbb{P}\_0(x)+\int\_{\mathcal{Y}} \beta(y)d\mathbb{P}\_1(y),$$
> and consider the following maximin reformulation of OT with the quadratic cost [1, Eq.4]:
> $$
> \sup_{\beta}\inf_{H}\widetilde{\mathcal{L}}(\beta,H) = \inf\_{T\sharp \mathbb{P}\_0 = \mathbb{P}\_1} \int\_{\mathcal{X}} ||x - T(x)||^2 d\mathbb{P}\_0(x) = \mathcal{L}^{*}.
> $$
>
> *Step 2 (Solution of the inner problem is always an OT map).*
>
> It can be shown that the minimizer of the inner problem exists ($H^{\beta}$), i.e.:
> $$
> H^{\beta}\in \text{arg}\min\_{H\sharp\mathbb{P}_0=\mathbb{P}'}\int\_{\mathcal{X}} \big\lbrace\|x-H(x)\|^{2}- \beta(H(x))\big\rbrace d\mathbb{P}\_0(x)
> $$
> Moreover, $H^{\beta}$ is an OT map between $\mathbb{P}_0$ and $\mathbb{P}'\stackrel{def}{=}H^{\beta}\sharp \mathbb{P}_0$.
>
> *Step 3 (Equivalence for inner objective values).*
>
> Since $H^{\beta}$ is the OT map between $\mathbb{P}\_0,\mathbb{P}' $, it can be represented as an ODE with zero acceleration ($\frac{df^{\beta}(x(t), t)}{dt}=0$) solution $T_{f^{\beta}}$ to the Benamour Brenier problem between $\mathbb{P}\_0,\mathbb{P}'$, for which $\|x-H^{\beta}(x)\|^{2}=\int_{0}^1 ||f^{\beta}(X_t, t)||^2 dt$, i.e.:
>
> $$\inf_{H}\tilde{\mathcal{L}}(\beta,H) = \inf_{T_f}\mathcal{L}(\beta, T_f).$$
>
> *Step 4 (Equivalence of the saddle point objective).*
>
> Take $\sup$ over $\beta\in\mathcal{C}\_{b,2}(\mathcal{Y})$ and get the final equivalence:
> $$\sup\_{\beta}\inf\_{H}\widetilde{\mathcal{L}}(\beta,H)= \sup\_{\beta}\inf\_{H}\mathcal{L}(\beta,T_{f})= \mathcal{L}^{*}.$$
>
> We cannot give the full proof due to the length limit of the answer, but per request, we can provide it. **At the same time, our paper focuses on solving the problem with $\epsilon > 0$.**
>
> **(2) Comparision with [1].**
>
> We are aware of [1]. However, since they address the unregularized OT problem, we have not compared our results to theirs. As per your request, we have trained our method with $\epsilon=0$ using the same image benchmark setup [2] and present our results alongside the results from Table 2 of [1] **in Table 4 of the attached pdf**. Since the method presented in [1] compares with the MMR method from [2], we also present its results.
>
> As we can see, ENOT with $\epsilon=0$ works better than the MM-R solver but slightly underperforms compared to [1].
>
> **(3) Comparison with [3].**
>
> We do not compare with DiffSB since the authors do not consider unpaired translation for the spaces of images larger than grayscale $32$x$32$. In their official GitHub repository there is no config for unpaired translation at all. Moreover, the authors themselves used not the Wiener prior for SB, see [3, Appendix J.3]. After trying to tune hyperparameters which we used for the Colored MNIST setup on our own and obtaining poor results, we decided not to scale this algorithm further to unpaired Celeba setup.
>
> **(4) Including dual form in Sec 2.**
>
> We agree that dual form formulation of OT problems also could be discussed in Section 2 since our proofs are based on it. We will add a discussion about dual OT formulation and methods based on it (such as [1] and [2]) to the main text of the final version.
>
> **(5) Comparison of SDE and ODE.**
>
> Our proposed algorithm has a larger gradient variance when $\epsilon$ is larger, which may affect the final quality. To improve the results, one can use more steps for sampling from SDE or adjust the learning rate. In the paper, we present results for different $\epsilon$ while keeping all the other hyperparameters the same.
>
> **(6) why is there a "1/|f|" term in Alg 1, when computing the KL? And what's $f_{n,m}$? I understand tht $f_n$ is the drift output at time step $n$.**
>
> We use $f_{n, m}$ to denote drift output at time step $n$ for the $m$-th object of the input sample batch. We use the average of the drifts as an estimation of $\int_{0}^{1} ||f(X_t, t)||^2 dt$ in the training objective.
>
> **Concluding remarks.**
> We would be grateful if you could let us know if the explanations we gave have been satisfactory in addressing your concerns about our work. If so, we kindly ask that you consider increasing your rating. We are also open to discussing any other questions you may have.
>
> **Additional references.**
>
> [1] Amos,. "On amortizing convex conjugates for optimal transport."
>
> [2] Korotin, et al. "Do neural optimal transport solvers work? a continuous wasserstein-2 benchmark."
>
> [3] De Bortoli, et al. "Diffusion Schrödinger bridge with applications to score-based generative modeling."

---

> > ### Comment · Reviewer_KyaP · 2023-08-17
> >
> > I thank the authors for the detailed response. It would be great to add these clarifications into the revision. I'll update my score.

---

### Official Review · Reviewer_WYHi · 2023-07-04

**Soundness:** 3 good
**Presentation:** 4 excellent
**Contribution:** 3 good
**Rating:** 9
**Confidence:** 4

**Summary:**

Inspired by how Sinkhorn duals are derived the authors adapt said derivation to the path measure and via the disintegration theorem they derive a novel unconstrained min-max objective for solving the Schrodinger bridge problem, the authors then proceed to showcase their method in eOT based tasks, introducing a new gaussian benchmark and displaying competitive results to previous approaches.  Additionally, the authors quantify errors in sampling and transport of approximate minimisers to their proposed schemes.


**Strengths:**

1. The extension of the Sinkhorn dual to the dynamic setting is rather elegant and certainly novel in the way it is carried out
2. The paper is excellent on the presentation side in regards to technical ideas, whilst the contributions are novel/creative they are presented in such a way that understanding them was not difficult.
3. The new formulation allows for a novel duality gap analysis, which is one of the few works analysing learned SBP methods in the approximate setting.
4. From a purely conceptual viewpoint the work is great and rather complete, just some clarifications/additions on the experimental side and motivations could be enhanced.

**Weaknesses:**

Outside of a concern in how different methods are compared (detailed in the questions), this paper is overall well written and has mostly sound experiments.

From the method standpoint, there are several potential weaknesses and lacking ablations which I will detail in the limitations.


**Questions:**

1. Some of the methods you compare to are closed form / non-iterative in the way they solve the half bridges e.g. [1] using GPs.  A thing to note is iterative approaches such as SGD can keep resampling from P_0 and P_1 in the toy tasks nonstop, effectively making the dataset set size somewhat proportional to the iterations, this strongly benefits generalisation capabilities, especially in higher dimensions in contrast to a the GP approach in [1] which is likely fitted on a small dataset. If we assume that the dataset is fixed across epochs K_f * batchsize would give a proxy as to the size of the dataset used in the DL approaches , from inspecting the code provided in the supplementary zip in particular the high dim Gaussians and toy examples IPYNB this quantity seems to be above 5000, whilst I suspect for [1] it might be lower. Even then I suspect you resample from the toy distributions (Gaussians) every time as indicated in your pseudocode in which case the dataset sizes are simply not comparable at all and the non-iterative approaches like MLE-SB are prone to be affected significantly by generalisation error (which scales slowly in number of samples for kernel methods, which also don't overcome the curse of dim as well as neural networks)  in particular in high dimensions.
2. Are the same number of timesteps used across the DL and the GP approaches when training ? computationally it feels unlikely the gram matrix would fit in memory for the given timesteps used with the DL approaches.
3. A suggestion here would be to compare all approaches in an additional table (e.g. in the appendix) under budgets in the number of steps and number of samples (and fairly ensure these are the same across methods), this can be useful to quantify the data efficiency of each approach. Furthermore, the settings under which [1] was run (steps and dataset size) should be reported.


**Limitations:**

Finding saddlepoints / solving min-max problems is typical quite a challenging task and was often a huge challenge in the training stability of GANs (very high variance training results). In contrast to IPF which is much nicer from an objective viewpoint (sequence of regression losses), this approach poses the question of how stable training is.

As none of the experiments have error bars (on training runs) it is difficult to see if the proposed approach is robust and “easy” to train.  I would suggest to the authors provide such results and in addition maybe comparisons to training loss with Chen 2021 joint or De Bortoli 2021.

---

> ### Author Rebuttal · Authors · 2023-08-09
>
> Dear Reviewer WYHi, Thank you for your comments. Here are the answers to your questions.
>
> **(1) Comparison with MLE-SB.**
>
> Comparing entropic OT methods is difficult because they are based on different principles: IPF-based (MLE-SB, DiffSB, FB-SDE), dual form based (LSOT, SCONES), semi-dual form based (ENOT, ours), and each of them has different hyperparameters that need to be tuned. **It seems like our paper is unique in the field which performs such a comprehensive comparison.** We have taken the most similar hyperparameters to those from the authors' repositories. In our initial results, all the methods (except LSOT and MLE-SB) work well and give good results. LSOT performs poorly because it only learns the barycentric projection, not the EOT plan. At the same time, it was not very clear to us why the results for MLE-SB were bad.
>
> It appears that [1] is the paper in which MLE-SB was first introduced, as it was the only one that used GP for training. Indeed, to ensure a fair comparison of the methods, rather than the way of solving regression problems, it would be better to use a neural network for all methods. This is because GP can be challenging to scale when used for setups with many time steps ($200$ for high-dimensional Gaussians) and a moderate amount of samples.
>
> We reran the experiments for MLE-SB using neural network parametrization of the drift function (as well as for other methods) and using $200$ time steps as we used for all SB methods. We use $1000$ IPF iterations and sample $512$ samples from each distribution $\mathbb{P}_0$ and $\mathbb{P}_1$ at each IPF iteration. **The updated results are shown in Tables 1, 2, and 3 of the attached PDF file.**
>
> As we can see, MLE-SB performs similarly or better than the other IPF-based method (DiffSB) and can also solve the problem in this setup. Now all the considered methods that learn the EOT plan work well on the Gaussian setup, and the goal of this experiment is achieved. The small residual error of all methods seems to be related more to hyperparameter tuning than to the nature of the algorithm used.
>
> **(2) Stability of the proposed method.**
>
> We run our method on the Gaussian setup five times and provide the means and standard deviations in **Tables 1, 2, and 3 of the attached PDF file**. In Figure 1, we also provide the plot of $\text{BW}_2^2\text{-UVP}$ (\%) between the ground truth EOT plan $\pi^{*}$ and the learned plan $\pi$ of ENOT and MLE-SB during training for $DIM=128$. Note that the steps on the plots represent different values due to the different nature of the algorithms. There are IPF steps for MLE-SB and outer problem steps for ENOT (ours). As we can see, both methods stably converge.
>
> **Concluding remarks**.
> We would be grateful if you could let us know if the explanations we gave have been satisfactory in addressing your concerns about our work. If so, we kindly ask that you consider increasing your rating. We are also open to discussing any other questions you may have.
>
> **Additional references.**
>
> [1] Vargas, Francisco, et al. "Solving schrödinger bridges via maximum likelihood." Entropy 23.9 (2021): 1134.

---

> > ### Comment · Reviewer_WYHi · 2023-08-10
> > **Very satisfied with rebuttal and the uploaded comparisons.**
> >
> > Dear Authors,
> >
> > I have read the rebuttal and can comment that you have clarified and resolved all the issues I raised. In particular, the stability plots and the further fair comparisons brought everything to a very clean conclusion. Additionally, I have also gone over the other reviews and the provided responses, overall it looks pretty satisfactory, the non-trivial prior results were quite a nice addition and comment to the robustness/flexibility of the proposed approach.
> >
> > I will increase my score accordingly once the platform allows such, at the time being it seems OpenReview is not allowing this until the discussion period is over, I do agree this work does seem to be the most thorough comparison yet for SBP solvers within the eOT context making this paper a substantial contribution beyond the novel approach that is proposed (which is also a solid and self-contained contribution on its own).

---

### Official Review · Reviewer_LQN6 · 2023-07-05

**Soundness:** 3 good
**Presentation:** 4 excellent
**Contribution:** 3 good
**Rating:** 9
**Confidence:** 3

**Summary:**

This paper focuses on neural optimal transport  and more particularly through a "dynamic schrodinger bridge" approach. I am not an expert in this particular topic, but I must say that the authors manage to make it quite readable and a good introduction to the methodology.

As far as I can tell, the particularity of the approach is to start from a known key constrained optimization problem in eq (11) that has been presented before and that is progressively explained. Then, departing from previous methods, the authors propose a Lagrangian approach, for which we now have an unconstrained problem and the need to train not only the drift function that allows sampling, but also some "beta network" (coined in as the beta network), in a min-max optimization approach, that is similar --but different -- from the usual GAN methodology.

After the rigorous theoretical treatment, the authors present some very nice experiment that seem to support their claims and the interesting features of their approach. I think the paper is quite challenging, but that it is also very stimulating and inspiring.

--
after rebuttal, I am still happy with the paper. Maintaining my score.

**Strengths:**

I feel slightly uncomfortable in assessing whether the proposed method is new or is not exactly or if the authors miss some particular recent and/or relevant method in their references, mostly because I cannot be considered an expert on the topic. Still, it looks to me that they are doing their best at providing a very objective account of the relevant literature and giving many pointers, so I am assuming they can be trusted when they present their contributions.

The proposed method seems quite ok to implement (cf Algorithm 1) and the results are very nice. I am not really aware of the appropriate metrics and usual evaluation criteria that should be used, but I felt compelled by the experiments, which made me want to try things out. I guess this is the most important aspect.

To summarize what I feel are the strengths of the paper:
- good theoretical overview, motivations and derivations
- the resulting method seems simple to implement
- interesting performances

**Weaknesses:**

The paper is a bit difficult to follow, but I don't think it should really be modified to be simpler. I guess it's mostly a matter of myself not being an expert in the topic.

Still, some changes here or clarifications here and there could help. I will mention them below.

**Questions:**

- p3 "Hence, one may optimize (5) over processes T for which T|x,y=W_xy for every x, y and set the last term in (6) to zero". How do you actually do that ? By enforcing gaussian dynamics ?
- the paragraph "continuous OT" reads a bit weak to me. All the references are just given in a row, without great care for actually discussing them and their connections with the proposed approach.

In algorithm 1, I have some questions.
- do you actually need to store the gradients when computing {X_n, f_n} for the computation of L_\beta ? It looks to me that Eul-Mar(X_0, T_{f_\theta}}) can actually run in some `no_grad` environment, or am I mistaken ? This would mean there is some possibility in strongly parallelizing this or even use some external workers to compute that ?

- On the contrary, in the inner loop (over k), you really need to store the gradients, right ? Is it necessary to use many inner iterations K_f ? Maybe I'm mistaken but I don't see that discussed, although this looks like a key computational burden when I have a look at algorithm 1, right ? Is it feasible to just record gradient for the last k steps ? for some of them ? Some thing that could help going faster ?

- I have no clue what this BW2-UVP metric is. Could you at least give us a hint that would help us avoid checking reference [25] ?

- You must check the references. Most of them are badly formated. You have "schr\"odinger" everywhere and you are missing many uppercases for proper nouns

**Limitations:**

I would say that the limitations are clearly mentioned

---

> ### Author Rebuttal · Authors · 2023-08-09
>
> Dear Reviewer LQN6, thank you for your comments. Here are the answers to your questions.
>
> **(1) p3 "Hence, one may optimize (5) over processes $T$ for which $T_{|x,y}=W_{|xy}$ for every $x, y$ and set the last term in (6) to zero". How do you actually do that ? By enforcing gaussian dynamics ?**
>
> We did not intend to do that in practice. And we do not do it in our algorithm. We just noticed that it can be considered as such a constrained optimisation. In fact, there seems to be no straightforward way to parameterise such a family of processes. However, as we noted in the main text (lines 76-78), if $T^*$ is a solution to SB with prior $W^{\epsilon}$ for any two distributions $\mathbb{P}\_0$ and $\mathbb{P}\_1$, then $T^*\_{|x,y} = W^{\epsilon}\_{|x,y}$  [28, Proposition 2.3].
>
> **(2) the paragraph "continuous OT" reads a bit weak to me. All the references are just given in a row, without great care for actually discussing them and their connections with the proposed approach.**
>
> Since many recent papers appeared on continuous OT, we focused the discussion only on the most relevant EOT and SB papers and only briefly mentioned the rest. Following your suggestion, we will extend the discussion of methods for solving other types of continuous OT (unregularized OT).
>
> **(3) do you actually need to store the gradients when computing ${X_n, f_n}$ for the computation of $L_\beta$ ? It looks to me that Eul-Mar($X_0$, $T_{f_\theta}$) can actually run in some no grad environment, or am I mistaken ? This would mean there is some possibility in strongly parallelizing this or even use some external workers to compute that ?**
>
> We do not have to store the gradients in the computation of ${X_n, f_n}$ for the computation of $L_\beta$ in Algorithm 1, and we indeed do not store them in our implementation. However, we have not yet tried to use computational schemes with external workers in our work. We appreciate the idea and will try it.
>
> **(4) On the contrary, in the inner loop (over k), you really need to store the gradients, right ? Is it necessary to use many inner iterations $K_f$ ? Maybe I'm mistaken but I don't see that discussed, although this looks like a key computational burden when I have a look at algorithm 1, right ? Is it feasible to just record gradient for the last k steps ? for some of them ? Some thing that could help going faster ?**
>
> We have to store them, but we can easily solve the memory problem using gradient checkpointing [1]. We have implemented it in our code. The hyperparameter $K_f$ affects the quality of the solution of the inner problem and can be tuned. We noticed that $K_f=1$ is too small, and the algorithm can easily diverge, but for $K_f=10$, the problem vanishes. We noticed in the limitations section that backpropagations through SDE may be computationally heavy. More efficient SDE solvers can be used to overcome this limitation.
>
> Another possibility is to consider approaches from stochastic optimal control or reinforcement learning, since with the fixed potential $\beta$, the inner problem can be formulated as a problem from these fields, as noted by **reviewer KyaP**. For example, one could consider $-||f(X_t, t)||^2\Delta t$ as a reward in an intermediate step and $\beta(X_{predicted})$ as an additional reward in the final step. In this case, one could use Q-learning or Actor-Critic approaches to learn from a segment of a trajectory without propagating through the whole trajectory.
>
> **(5) I have no clue what this BW2-UVP metric is. Could you at least give us a hint that would help us avoid checking reference [25] ?**
>
> It is the Wasserstein-2 distance between distributions $\mathbb{P}$ and $\mathbb{Q}$ that are coarsened to Gaussians and normalized by the variance of the distribution $\mathbb{Q}$:
> $$
> \text{B}\mathbb{W}\_{2}^{2}\text{-UVP}\big(\mathbb{P}, \mathbb{Q} \big) = \frac{100 \\%}{\frac{1}{2}\text{Var}(\mathbb{Q})} \mathbb{W}\_{2}^{2} (\mathcal{N}(\mu\_{\mathbb{P}}, \Sigma\_{\mathbb{P}}), \mathcal{N}(\mu\_{\mathbb{Q}}, \Sigma\_{\mathbb{Q}})).
> $$
>
> We will add this definition to the main text.
>
> **(6) You must check the references. Most of them are badly formated. You have "schr"odinger" everywhere and you are missing many uppercases for proper nouns.**
>
> Thanks so much for pointing out this. We will check and correct the references.
>
> **Concluding remarks.**
> We would be grateful if you could let us know if the explanations we gave have been satisfactory in addressing your concerns about our work. We are also open to discussing any other questions you may have.
>
> **Additional references.**
>
> [1] Chen, Tianqi, et al. "Training deep nets with sublinear memory cost." arXiv preprint arXiv:1604.06174 (2016).
> [2] Christian Léonard. A survey of the schr\" odinger problem and some of its connections with
> optimal transport.

---

### Official Review · Reviewer_TSgQ · 2023-07-06

**Soundness:** 4 excellent
**Presentation:** 4 excellent
**Contribution:** 4 excellent
**Rating:** 8
**Confidence:** 3

**Summary:**

The main idea of the paper is to estimate a stochastic map for the entropic optimal transport problem using its connection to the dynamic Schrödinger bridge (SDB) problem. The authors formulate the SDB as a saddle point problem of an associated Lagrangian. Then they recover the transport plan as the joint distribution of the solution to the dynamic Schrödinger bridge problem's initial and final values, while the transport is encoded in the drift term of the learned stochastic process that is the solution of the SDB. Compared to previous methods, the method at hand offers more stability for small entropic regularization coefficients. The errors on the drift term solution and on the transport map are quantified given the corresponding duality gaps of the inner and outer optimization problems of the saddle point objective. Finally, the approach is supported by experimental evaluation.

**Strengths:**

The paper is extremely well-written and reader-friendly.

- Several remarks are made to facilitate reading and to provide intuition about technical notions.
- A guarantee on the quality of the saddle point solution is provided.
- Addressing the small $\epsilon$ case, which is a source of instability of several other methods.

**Weaknesses:**

* In line 223, it is mentioned that the negative entropy is not strongly convex. This is false as the function $p\mapsto x\ln(x)$ has second derivate $x \mapsto \frac{1}{x}$ which is bounded from below by $1$ on the interval $[0,1]$. See for example Section 4.1 of reference [2]. As a result, the comparison to [1]  (reference [5] in the paper) needs to be reconsidered.

* There is no concluding section.

* Some experimental section metrics are not introduced.

  * The $\rm{BW_2^2-UVP}$ is not introduced even in the appendix, although a reference for it is given.

  * FID: the previous remark applies.

* The parametrization $g(X_t,t) = X_t + f(X_t,t)\Delta_t $ should have been indicated in the main paper rather than in the appendix as although it is mathematically equivalent to the parametrization presented in the main paper, it allowed better results on the CelebA dataset according to the appendix

  ## Minor remarks

* Problem with links: For some reason, the bibliographic references links along with links to equations and sections etc. are not working.

* $\pi^{W^\epsilon}$ is introduced for the first time in Equation (8) without being defined.

* $W_{|x,y}$ is not explicitly defined, although one can infer its meaning from the definition of $T_{|x,y}$.

* Theorem 4.1: I think it should be "every pair $(\beta^*,T_{f^*})$ for (13)" rather than "for (12)" as problem (13) is a saddle point problem.

* Reference to Algorithm 2: in line 195, Algorithm 2 is referenced. However, it is not indicated that it is written in the appendix.

* Suggestion: index $m$ can be removed in the "$\widehat{KL} \leftarrow$" line of Algorithm 1  since the sum terms are already indicated to be the values of $f_n$, or it is possible to indicate $\sum_{m=1}^{|f_n|}$.

* Line 158: I think it should be added "that is bounded from above" to "a continuous function".

## References

[1] Asadulaev, A., Korotin, A., Egiazarian, V., & Burnaev, E. (2022). Neural optimal transport with general cost functionals. *arXiv preprint arXiv:2205.15403*.

[2] Peyré, G., & Cuturi, M. (2019). Computational optimal transport: With applications to data science. *Foundations and Trends® in Machine Learning*, *11*(5-6), 355-607.

**Questions:**

* Computation of $\mathbb{E}\left[\int_0^1\Vert f(X_t,t)\Vert^2{\rm d}t\right]$ : in line 197, it is indicated that the mean of the $f(x,t)$ is used. Is this justified by the Riemann integral discrete approximation? If so, does a trapezoidal rule for approximating the integral improve the result?
* Is it straightforward to generalize the approach to costs other than the squared Euclidean distance?  Does it fundamentally change the nature of the associated stochastic process
* Did the authors try to apply the method to the domain adaptation (DA) problem as several DA methods rely on optimal transport ?
----------
I have read the authors's rebuttal. They have addressed my concerns.

**Limitations:**

The impact of the paper and the limitations of the contribution are clearly discussed in Section 6.

---

> ### Author Rebuttal · Authors · 2023-08-09
>
> Dear Reviewer TSgQ, thank you for your comments. Here are the answers to your questions.
>
> **(1) In line 223, it is mentioned that the negative entropy is not strongly convex. This is false as the function $p\mapsto x\ln(x)$ has second derivate $x \mapsto \frac{1}{x}$ which is bounded from below by $1$ on the interval $[0,1]$. See for example Section 4.1 of reference [2]. As a result, the comparison to [1] (reference [5] in the paper) needs to be reconsidered.**
>
> Recall that the negative (differential) entropy of a distribution with density $p(x)$ is given by $-H(p) = \int p(x)\log p(x)dx$. The negative entropy is indeed $\frac{1}{M}$-strongly convex if we only consider distribution $p$ whose density $p(x)$ is bounded by a constant $M>0$ (this follows from your argument). However, we work with general continuous distributions ($\mathbb{P}_0,\mathbb{P}_1,\widehat{\pi}$, etc.) which may not satisfy this assumption. For example, $\mathcal{N}(x|0,\sigma)$ for small $\sigma$ may have density at $0$, which is greater than any given constant $M$. Therefore, the differential entropy is not $\frac{1}{M}$-strongly convex for any $M>0$.
>
> **(2) Some experimental section metrics are not introduced.**
>
> *We will add the explanations and definitions of the metrics to Appendix.*
>
> **(3) The parametrization $g(X_t,t) = X_t + f(X_t,t)\Delta_t $ should have been indicated in the main paper rather than in the appendix as although it is mathematically equivalent to the parametrization presented in the main paper, it allowed better results on the CelebA dataset according to the appendix.**
>
> *We will include a comment regarding the parametrization utilized in the primary text (to Section 4.2).*
>
> **(4) Problem with links: For some reason, the bibliographic references links along with links to equations and sections etc. are not working.**
>
> Thank you for pointing this out. We will fix it.
>
> **(5) Minor remarks.**
>
> Thank you for your comments. We will fix all the issues and think it will further improve the clarity.
>
> **(6) Computation of $\mathbb{E}\left[\int_0^1\Vert f(X_t,t)\Vert^2{\rm d}t\right]$ in line 197, it is indicated that the mean of the
>  is used. Is this justified by the Riemann integral discrete approximation? If so, does a trapezoidal rule for approximating the integral improve the result?**
>
> Yes, we used the mean as a discrete approximation of $\mathbb{E}\left[\int_0^1\Vert f(X_t,t)\Vert^2{\rm d}t\right]$. We have not tried to use other types of approximation of this integral that might improve the proposed algorithm. Thank you for your suggestion.
>
> **(7) Is it straightforward to generalize the approach to costs other than the squared Euclidean distance? Does it fundamentally change the nature of the associated stochastic process.**
>
> Yes, it is quite straightforward. We focus only on EOT with the quadratic cost $c(x,y)=\frac{1}{2}\|x-y\|^{2}$ which coincides with SB with the Wiener prior $W^{\epsilon}$. However, one could use a different prior $Q_v$, given by the SDE:
> $$
> Q_v: dX_t = v(X_t, t)dt + \sqrt{\epsilon}dW_t,
> $$
> and solve the problem
>
> $$
> \inf_{T_f \in \mathcal{D}(\mathbb{P}_0, \mathbb{P}_1)} \text{KL}(T_f || Q_v) = \inf\_{T\_f \in \mathcal{D}(\mathbb{P}\_0, \mathbb{P}\_1)} \frac{1}{2\epsilon} \mathbb{E}\_{T\_f}[\int\_{0}^1 ||f(X_t, t) - v(X_t, t)||^2 dt].
> $$
>
> Here we just use the known expression to $\text{KL}(T_f|| Q_v)$ between two diffusion processes through their drift functions as we did in our paper. Using the same derivation as in our paper, it can be shown that this new problem is equivalent to solving the EOT with cost $c(x,y) = -\log \pi^{Q_v}(y|x)$, where $\pi^{Q_v}(y|x)$ is a conditional distribution of the stochastic process $Q_{v}$ at time $t=1$ given the starting point $x$ at time $t=0$. For example, for $W^{\epsilon}$ (which we used) we have $c(x,y) = -\log \pi^{W^{\epsilon}}(y|x) = \frac{1}{2 \epsilon}(y-x)^T(y-x) + \text{Const}$, i.e., we get the quadratic cost. Thus, using different priors for the Schrodinger bridge problem makes it possible to solve Entropic OT for other costs.
>
> It seems that all our proofs can be extended to any prior process $Q_v$ just by slightly changing the minimax functional:
>
> $$
> \sup\_{\beta} \inf\_{T\_{f}} (\frac{1}{2\epsilon} \mathbb{E}\_{T\_f}[\int\_{0}^1 ||f(X\_t, t) - v(X\_t, t)||^2 dt] + \int\_{\mathcal{Y}} \beta\_{\phi}(y) d\mathbb{P}\_1(y) - \int\_{\mathcal{Y}} \beta\_{\phi}(y) d\pi\_1^{T\_f}(y)).
> $$
>
> We conducted a toy experiment to support this claim and consider $Q_v$ with $\epsilon=0.01$ and $v(x, t) = \nabla \log p(x)$, where $\log p(x)$ is a 2D distribution with a wave shape, **see figure 2 of the attached pdf.** Intuitively, it means that trajectories will be concentrated in the regions with a high density of $p$. In Figure 2, There the grey-scale color map represents the density of $p$, start points ($\mathbb{P}\_0$) are green, target points ($\mathbb{P}\_1$) are red, obtained trajectories are pink and mapped points are blue.
>
> **(8) Did the authors try to apply the method to the domain adaptation (DA) problem as several DA methods rely on optimal transport?**
>
> No, we did not try.
>
> **Concluding remarks.**
> We would be grateful if you could let us know if the explanations we gave have been satisfactory in addressing your concerns about our work. If so, we kindly ask that you consider increasing your rating. We are also open to discussing any other questions you may have.
>
> **Additional references.**
>
> [1] Asadulaev, A., Korotin, A., Egiazarian, V., \& Burnaev, E. (2022). Neural optimal transport with general cost functionals. arXiv preprint arXiv:2205.15403.
>
> [2] Michele Pavon and Anton Wakolbinger. On free energy, stochastic control, and schrödinger
> processes. In Modeling, Estimation and Control of Systems with Uncertainty: Proceedings of a
> Conference held in Sopron, Hungary, September 1990, pages 334–348. Springer, 1991.

---

> > ### Comment · Reviewer_TSgQ · 2023-08-14
> > **Thanks for the rebuttal**
> >
> > Dear Authors,
> >
> > I would like to thank you for the detailed and very well written rebuttal. It addressed all of my concerns. I update my score.

---

### Author Rebuttal · Authors · 2023-08-09

Dear reviewers, thank you for taking the time to review our paper.

Your valuable feedback and constructive comments are greatly appreciated. We are particularly pleased that all reviewers found our paper well-written and easy to read (TSgQ, LQN6, WYHi, KyaP). We are also pleased that you find our duality gap analysis and guarantee of the quality of the saddle point solution important (TSgQ, WYHi), that our experiments are extensive and include many related baselines (KyaP), and that the whole work is great and quite complete (WYHi).

Please, find the answers to your questions below. **Please note that we have added tables and figures in the attached pdf to support our responses to the reviewers WYHi, KyaP, and TSgQ.**

---

### Decision · Program_Chairs · 2023-09-21

**Decision:**

Accept (oral)

**Comment:**

The paper proposes a novel algorithm for estimating the entropic optimal transport by
bridging the gap with the Schrödinger bridge problem. The authors propose a saddle-point, maxmin optimization problel
that allows to train the neural OT.

All reviewers agree that the paper proposes a novel approaches with sound theoretical arguments and
supported by empirical evidences.